# Alleviating Adversarial Attacks on Variational Autoencoders with MCMC

**Anna Kuzina**
Vrije Universiteit Amsterdam
`a.kuzina@vu.nl`

**Max Welling**
Universiteit van Amsterdam
`m.welling@uva.nl`

**Jakub M. Tomczak**
Vrije Universiteit Amsterdam
`j.m.tomczak@vu.nl`

## Abstract

Variational autoencoders (VAEs) are latent variable models that can generate complex objects and provide meaningful latent representations. Moreover, they could be further used in downstream tasks such as classification. As previous work has shown, one can easily fool VAEs to produce unexpected latent representations and reconstructions for a visually slightly modified input. Here, we examine several objective functions for adversarial attack construction proposed previously and present a solution to alleviate the effect of these attacks. Our method utilizes the Markov Chain Monte Carlo (MCMC) technique in the inference step that we motivate with a theoretical analysis. Thus, we do not incorporate any extra costs during training, and the performance on non-attacked inputs is not decreased. We validate our approach on a variety of datasets (MNIST, Fashion MNIST, Color MNIST, CelebA) and VAE configurations ($\beta$-VAE, NVAE, $\beta$-TCVAE), and show that our approach consistently improves the model robustness to adversarial attacks.

## 1 Introduction

Variational Autoencoders (VAEs) [27, 34] are latent variable models parameterized by deep neural networks and trained with variational inference. Recently, it has been shown that VAEs with hierarchical structures of latent variables [33], coupled with skip-connections [30, 37], can generate high-quality images [15, 39]. An interesting trait of VAEs is that they allow learning meaningful latent space that could be further used in downstream tasks [7, 22]. These successes of VAEs motivate us to explore the *robustness* of the resulting latent representations to better understand the capabilities and potential vulnerabilities of VAEs. Here, we focus on *adversarial attacks* on VAEs to verify robustness of latent representations that is especially important in such applications as anomaly detection [1, 30] or data compression [5, 20].

The main questions about adversarial attacks for VAEs are mainly focused on how they could be formulated and alleviated. In [18], it is proposed to minimize the KL-divergence between an adversarial input and a target input to learn an adversarial attack for the vanilla VAE. Further, in [28], it is shown that a similar strategy can be used to attack hierarchical VAEs. To counteract the adversarial attacks, the authors of [42] suggest using a modified VAE objective, namely, $\beta$-TCVAE, that increases VAE robustness, especially when coupled with a hierarchical structure. It was shown in [10] that $\beta$-VAEs tend to be more robust to adversarial attacks in terms of the $r$-metric proposed therein. The authors of [6] presented that the adversarial robustness can be achieved by constraining the Lipschitz constant of the encoder and the decoder. [12, 13] introduced modifications in the VAE framework that allow for better robustness against the adversarial attacks on downstream classification tasks. In our work, we consider $\beta$-VAE, $\beta$-TCVAE and a hierarchical VAE, and outline a defence strategy that improves robustness to attacks on the encoder and the downstream classification task. The proposed method is applied during inference and, therefore, can be combined with other known techniques to get more robust latent representations.

36th Conference on Neural Information Processing Systems (NeurIPS 2022).

An adversarial attack on a VAE is usually formulated as an additive perturbation $\varepsilon$ of the real data point $\mathbf{x}^r$ so that the resulting point is perceived by a model as if it is a totally different image (either during reconstruction or in the downstream classification task) [18]. In Figure 1, we depict an example of an attack on the encoder. The reference point $\mathbf{x}^r$ and the adversarial point $\mathbf{x}^a$ are almost indistinguishable, but they are encoded into different regions in the latent space. As a result, their reconstructions also differ significantly.

In this paper, we propose the method motivated by the following hypothesis: *An adversarial attack maps the input to a latent region with a lower probability mass assigned by the true posterior (proportional to the conditional likelihood times the marginal over latents) and, eventually, we obtain incorrect reconstructions*. Therefore, a potential manner to alleviate the effect of an attack may rely on running a Markov chain to move the latent representation back to a more probable latent region. Such a defence is reasonable because we do not modify the training procedure or the model itself, we only insert a correction procedure. As a result, we propose to counteract adversarial attacks by enhancing the variational inference with Markov Chain

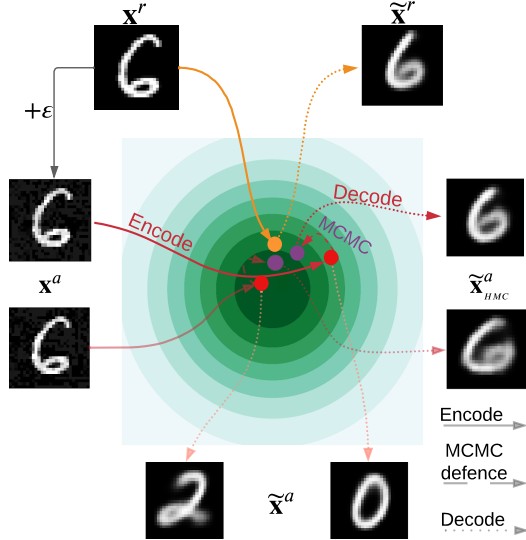

Figure 1: An example of an unsupervised encoder attack on VAE with 2D latent space and the proposed defence. Given a single reference point $\mathbf{x}^r$ we learn additive perturbation $\varepsilon$, s.t. perturbed input $\mathbf{x}^a$ has the most different latent code and, therefore, the reconstruction $\widetilde{\mathbf{x}}^a$. We observe that a single reference point can be mapped to extremely different regions of the latent space but using MCMC we are able to move them closer to the initial position so that the reconstruction $\widetilde{\mathbf{x}}^a_{HMC}$ is similar to the initial one $\widetilde{\mathbf{x}}^r$.

Monte Carlo (MCMC) sampling. The illustrative example depicted in the Figure 1 shows that the latent code of the adversarial input (red circle) moves closer to the latent code of the reference point (orange circle) after applying the MCMC (purple circle).

The contribution of this work is the following:

- We propose to use an MCMC technique during inference to correct adversarial attacks on VAEs.

- We show theoretically that the application of an MCMC technique could indeed help to counteract adversarial attacks (Theorem 1).

- We indicate empirically that the previously proposed strategies to counteract adversarial attacks do not generalize well across various datasets.

- We show empirically that the proposed approach (i.e., a VAE with an MCMC during inference) outperforms all baselines by a significant margin.

## 2 Background

### 2.1 Variational Autoencoders

Let us consider a vector of observable random variables, $\mathbf{x} \in \mathcal{X}^D$ (e.g., $\mathcal{X} = \mathbb{R}$) sampled from the empirical distribution $p_e(\mathbf{x})$, and vectors of latent variables $\mathbf{z}_k \in \mathbb{R}^{M_k}$, $k = 1, 2, \ldots, K$, where $M_k$ is the dimensionality of each latent vector. First, we focus on a model with $K = 1$ and the joint distribution $p_\theta(\mathbf{x}, \mathbf{z}) = p_\theta(\mathbf{x}|\mathbf{z})p(\mathbf{z})$. The marginal likelihood is then equal to $p_\theta(\mathbf{x}) = \int p_\theta(\mathbf{x}, \mathbf{z})\mathrm{d}\mathbf{z}$. VAEs exploit variational inference [25] with a family of variational posteriors $\{q_\phi(\mathbf{z}|\mathbf{x})\}$, also referred to as encoders, that results in a tractable objective function, i.e., the Evidence Lower BOund (ELBO): $\mathcal{L}(\phi, \theta) = \mathbb{E}_{p_e(\mathbf{x})} \left( \mathbb{E}_{q_\phi(\mathbf{z}|\mathbf{x})} \ln p_\theta(\mathbf{x}|\mathbf{z}) - D_{\mathrm{KL}}\left[ q_\phi(\mathbf{z}|\mathbf{x}) \| p(\mathbf{z}) \right] \right)$.

$\beta$-VAE [21] uses a modified objective by weighting the $D_{\mathrm{KL}}$ term by $\beta > 0$. In the case of $K > 1$, we consider a hierarchical latent structure with the generative model of the following form: $p_\theta(\mathbf{x}, \mathbf{z}_1, \ldots, \mathbf{z}_K) = p_\theta(\mathbf{x}|\mathbf{z}_1, \ldots, \mathbf{z}_K) \prod_{k=1}^{K} p_\theta(\mathbf{z}_k|\mathbf{z}_{>k})$. There are various possible formulations

Table 1: Different types of attacks on the VAE. We denote $g_\theta(z)$ the deterministic mapping induced by decoder $p_\theta(x|z)$ and as $p_\psi(y|z)$ classification model in the latent space (downstream task). [*] Only used during VAE training

|  | REFERENCE | $f(x)$ | $\Delta[A, B]$ | $\|\cdot\|_p$ | TYPE |
|---|---|---|---|---|---|
| Latent Space Attack | [6, 18, 42] | $q_\phi(\cdot|x)$ | $D_{\mathrm{KL}}[A\|B]$ | 2 | Supervised |
| Unsupervised Encoder Attack | [28] | $q_\phi(\cdot|x)$ | $\mathrm{SKL}[A\|B]$ | 2 | Unsupervised |
| Targeted Output Attack | [18] | $g_\theta(\tilde{z}), \tilde{z} \sim q_\phi(\cdot|x)$ | $\|A - B\|_2^2$ | 2 | Supervised |
| Maximum Damage Attack | [6, 10] | $g_\theta(\tilde{z}), \tilde{z} \sim q_\phi(\cdot|x)$ | $\|A - B\|_2^2$ | 2 | Unsupervised |
| Projected Gradient Descent Attack[*] | [13] | $q_\phi(\cdot|x)$ | $\mathcal{WD}[A, B]$ | inf | Unsupervised |
| Adversarial Accuracy | [12, 13] | $p_\psi(y|\tilde{z}), \tilde{z} \sim q_\phi(\cdot|x)$ | CROSS ENTROPY | inf | Unsupervised |

of the family of variational posteriors. However, here we follow the proposition of [37] with the autoregressive inference model, namely, $q_\phi(\mathbf{z}_1, \ldots, \mathbf{z}_K|\mathbf{x}) = q_\phi(\mathbf{z}_K|\mathbf{x}) \prod_{k=1}^{K-1} q_{\theta,\phi}(\mathbf{z}_k|\mathbf{z}_{>k}, \mathbf{x})$. This formulation was used, among others, in NVAE [39]. Because of the top-down structure, it allows sharing data-dependent information between the inference model and the generative model.

## 2.2 Adversarial attacks

An *adversarial attack* is a slightly deformed data point that results in an undesired or unpredictable performance of a model [19]. In this work, we focus on the attacks that are constructed as an additive perturbation of the real data point $\mathbf{x}^r$ (which we will refer to as *reference*), namely:

$$\mathbf{x}^a = \mathbf{x}^r + \varepsilon, \text{ where} \tag{1}$$

$$\|\varepsilon\|_p \leq \delta, \tag{2}$$

where $\delta$ is the radius of the attack. The additive perturbation $\varepsilon$ is chosen in such a way that the attacked point does not differ from the reference point too much in a sense of a given similarity measure. It is a solution to an optimization problem solved by the attacker. The optimization problem could be formulated in various manners by optimizing different objectives and by having specific constraints and/or assumptions.

**Attack construction** Let $f(x)$ be part of the model available to the attacker. In the case of VAEs, this, for example, may be an encoder network or an encoder with the downstream classifier in the latent space. The attacker uses a similarity measure $\Delta$ to learn an additive perturbation to a reference point. We consider two settings: *unsupervised* and *supervised*. In the former case, the perturbation is supposed to incur the largest possible change in $f$:

$$\varepsilon = \arg \max_{\|\varepsilon\|_p \leq \delta} \Delta\left[f(\mathbf{x}^r + \varepsilon), f(\mathbf{x}^r)\right]. \tag{3}$$

The latter setting requires a *target point* $\mathbf{x}^t$. The perturbation attempts to match the output for the target and the attacked points, namely:

$$\varepsilon = \arg \min_{\|\varepsilon\|_p \leq \delta} \Delta\left[f(\mathbf{x}^r + \varepsilon), f(\mathbf{x}^t)\right]. \tag{4}$$

There are different ways to select $f$ and $\Delta$ in the literature. Furthermore, different $L_p$-norms can be used to restrict the radius of the attack. Table 1 summarizes recent work on the topic.

In this work, we focus on attacking the encoder and the downstream classification task using unsupervised adversarial attacks. In the former, we maximize the symmetric KL-divergence to get the point with the most unexpected latent code and, therefore, the reconstruction. In the latter, the latent code is passed to a classifier. The attack is trained to change the class of the point by maximizing the cross-entropy loss. However, the method we propose in Section 3 is not limited to these setups since it is agnostic to how the attack was trained.

**Robustness measures**   To measure the robustness of the VAE as well as the success of the proposed defence strategy, we focus on two metrics: MSSSIM and Adversarial accuracy.

For latent space attacks we follow [28] in using Multi-Scale Structural Similarity Index Measure (MSSSIM) [41]. We calculate $\mathrm{MSSSIM}[\widetilde{\mathbf{x}}^r, \widetilde{\mathbf{x}}^a]$, i.e., the similarity between reconstructions of $\mathbf{x}^r$ and the corresponding $\mathbf{x}^a$. We do not report the similarity between a reference and the corresponding adversarial input, since this value is the same for all the considered models (for a given attack radius). A successful adversarial attack would have a small value of $\mathrm{MSSSIM}[\widetilde{\mathbf{x}}^r, \widetilde{\mathbf{x}}^a]$.

For the attacks on the downstream classifier, we follow [12, 13] and calculate the adversarial accuracy. For a given trained VAE model, we first train a linear classifier using latent codes as features. Afterwards, the attack is trained to fool the classifier. Adversarial accuracy is the proportion of points for which the attack was unsuccessful. Namely, when the predicted class of the reference and corresponding adversarial point are the same.

## 3   Preventing adversarial attacks with MCMC

**Assumptions and the hypothesis**   We consider a scenario in which an attacker has access to the VAE encoder and, where relevant, to the downstream classification model. Above, we presented in detail how an attack could be performed. We assume that the defender cannot modify these components, but it is possible to add elements that the attacker has no access to.

We hypothesize that *the adversarial attacks result in incorrect reconstructions because the latent representation of the adversarial input lands in a region with a lower probability mass assigned by the true posterior.* This motivates us to use a method which brings the latents "back" to a highly probable region as a potential defence strategy. Since the adversarial attack destroys the input irreversibly, at the first sight it seems impossible to reconstruct the latent representation of the reference point. We aim at showing that this is possible to some degree theoretically and empirically (Appendix C.1).

**The proposed solution**   In order to steer the latents towards high probability regions, we propose to utilize an MCMC method during inference time. Since we are not allowed to modify the VAE or its learning procedure, this is a reasonable procedure from the defender's perspective. Another positive outcome of such an approach is that in a case of no attack, the latents given by the MCMC sampling will be closer to the mean of the posterior, thus, the reconstruction should be sharper or, at least, not worse. Note that the variational posterior approximates the true posterior from which the MCMC procedure samples. We propose to sample from $q^{(t)}(\mathbf{z}|\mathbf{x}) = \int Q^{(t)}(\mathbf{z}|\mathbf{z}_0) q_\phi(\mathbf{z}_0|\mathbf{x}) d\mathbf{z}_0$, where $Q^{(t)}(\mathbf{z}|\mathbf{z}_0)$ is a transition kernel of MCMC with $t$ steps and the target distribution $\pi(\mathbf{z}) = p_\theta(\mathbf{z}|\mathbf{x}) \propto p_\theta(\mathbf{x}|\mathbf{z})p(\mathbf{z})$. Alternatively, we can use optimization to find the mode of the posterior, however, MCMC can add extra benefits such as exploration of the typical set and randomness (see Appendix B.3 for details).

To further analyze the proposed approach, we start with showing the following lemma:

**Lemma 1** *Consider true posterior distributions of the latent code $\mathbf{z}$ for a data point $\mathbf{x}$ and its corrupted version $\mathbf{x}^a$. Assume also that $\ln p_\theta(\mathbf{z}|\mathbf{x})$ is twice differentiable over $\mathbf{x}$ with continuous derivatives at the neighbourhood around $\mathbf{x} = \mathbf{x}^r$. Then the KL-divergence between these two posteriors could be expressed using the small o notation of the radius of the attack, namely:*

$$D_{\mathrm{KL}}\left[p_\theta(\mathbf{z}|\mathbf{x}^r)\|p_\theta(\mathbf{z}|\mathbf{x}^a)\right] = o(\|\varepsilon\|). \tag{5}$$

*Proof* See Appendix A.

According to Lemma 1, the difference (in the sense of the Kullback-Leibler divergence) between the true posteriors for a data point $\mathbf{x}$ and its corrupted version $\mathbf{x}^a$ decreases faster than the norm of the attack radius. However, it is important to note that this result is only valid for asymptotically small attack radius.

Next, the crucial step to show is whether we can quantify somehow the difference between the distribution over latents for $\mathbf{x}^a$ after running MCMC with $t$ steps, $q^{(t)}(\mathbf{z}|\mathbf{x}^a)$, and the variational distribution for $\mathbf{x}^r$, $q_\phi(\mathbf{z}|\mathbf{x}^r)$. We provide an important upper-bound for the Total Variation distance (TV)[1] between $q^{(t)}(\mathbf{z}|\mathbf{x}^a)$ and $q_\phi(\mathbf{z}|\mathbf{x}^r)$ in the following lemma:

---
[1]The Total Variation fulfills the triangle inequality and it is a proper distance measure.

**Lemma 2** *The Total Variation distance (*TV*) between the variational posterior with MCMC for a given corrupted point* $\mathbf{x}^a$, $q^{(t)}(\mathbf{z}|\mathbf{x}^a)$, *and the variational posterior for a given data point* $\mathbf{x}^r$, $q_\phi(\mathbf{z}|\mathbf{x}^r)$, *can be upper bounded by the sum of the following three components:*

$$\mathrm{TV}\left[q^{(t)}(\mathbf{z}|\mathbf{x}^a), q_\phi(\mathbf{z}|\mathbf{x}^r)\right] \leq \mathrm{TV}\left[q^{(t)}(\mathbf{z}|\mathbf{x}^a), p_\theta(\mathbf{z}|\mathbf{x}^a)\right]$$
$$+ \sqrt{\tfrac{1}{2} D_{\mathrm{KL}}\left[p_\theta(\mathbf{z}|\mathbf{x}^r)\|p_\theta(\mathbf{z}|\mathbf{x}^a)\right]}$$
$$+ \sqrt{\tfrac{1}{2} D_{\mathrm{KL}}\left[q_\phi(\mathbf{z}|\mathbf{x}^r)\|p_\theta(\mathbf{z}|\mathbf{x}^r)\right]}. \qquad (6)$$

*Proof* See Appendix A.

The difference expressed by $\mathrm{TV}\left[q^{(t)}(\mathbf{z}|\mathbf{x}^a), q_\phi(\mathbf{z}|\mathbf{x}^r)\right]$ is thus upper-bounded by the following three components:

- The TV between $q^{(t)}(\mathbf{z}|\mathbf{x}^a)$ and the real posterior for the corrupted image, $p_\theta(\mathbf{z}|\mathbf{x}^a)$. Theoretically, if $t \to \infty$, $q^{(\infty)}(\mathbf{z}|\mathbf{x}^a) = p_\theta(\mathbf{z}|\mathbf{x}^a)$ and, hence, $\mathrm{TV}\left[q^{(\infty)}(\mathbf{z}|\mathbf{x}^a), p_\theta(\mathbf{z}|\mathbf{x}^a)\right] = 0$.
- The second component is the square root of the KL-divergence between the real posteriors for the image and its corrupted counterpart. Lemma 1 gives us information about this quantity.
- The last element, the square root of $D_{\mathrm{KL}}\left[q_\phi(\mathbf{z}|\mathbf{x}^r)\|p_\theta(\mathbf{z}|\mathbf{x}^r)\right]$, quantifies the *approximation gap* [16], i.e., the difference between the best variational posterior from a chosen family, and the true posterior. This quantity has no direct connection with adversarial attacks. However, as we can see, using a rich family of variational posteriors can help us to obtain a tighter upper-bound. In other words, taking flexible variational posteriors allows to counteract attacks. This finding is in line with the papers that propose to use hierarchical VAEs as the means for preventing adversarial attacks [42].

Eventually, by applying Lemma 1 to Lemma 2, we obtain the following result:

**Theorem 1** *The upper bound on the total variation distance between samples from MCMC for a given corrupted point* $\mathbf{x}^a$, $q^{(t)}(\mathbf{z}|\mathbf{x}^a)$, *and the variational posterior for the given real point* $\mathbf{x}^r$, $q_\phi(\mathbf{z}|\mathbf{x}^r)$, *is the following:*

$$\mathrm{TV}\left[q^{(t)}(\mathbf{z}|\mathbf{x}^a), q_\phi(\mathbf{z}|\mathbf{x}^r)\right] \leq \mathrm{TV}\left[q^{(t)}(\mathbf{z}|\mathbf{x}^a), p_\theta(\mathbf{z}|\mathbf{x}^a)\right]$$
$$+ \sqrt{\tfrac{1}{2} D_{\mathrm{KL}}\left[q_\phi(\mathbf{z}|\mathbf{x}^r)\|p_\theta(\mathbf{z}|\mathbf{x}^r)\right]}$$
$$+ o(\sqrt{\|\varepsilon\|}). \qquad (7)$$

*Proof* See Appendix A.

As discussed already, the first component gets smaller with more steps of the MCMC. The second component could be treated as a *bias* of the family of variational posteriors. Finally, there is the last element that corresponds to a constant error that is unavoidable. However, this term decays faster than the square root of the attack radius for the asymptotically small attack radius.

**Specific implementation of the proposed approach** In this paper, we use a specific MCMC method, namely, the Hamiltonian Monte Carlo (HMC) [8, 17]. Once the VAE is trained, the attacker calculates an adversarial point $\mathbf{x}^a$ using the encoder of the VAE. After the attack, the latent representation of $\mathbf{x}^a$ is calculated, $\mathbf{z}^a$, and used as the initialization of the HMC.

In the HMC, the target (unnormalized) distribution is $p(\mathbf{x}^a|\mathbf{z})p(\mathbf{z})$. The Hamiltonian is then the energy of the joint distribution of $\mathbf{z}$ and the auxilary variable $\mathbf{p}$, that is:

$$H(\mathbf{z}, \mathbf{p}) = U(\mathbf{z}) + K(\mathbf{p}), \qquad (8)$$
$$U(\mathbf{z}) = -\ln p_\theta(\mathbf{x}^a|\mathbf{z}) - \ln p(\mathbf{z}), \qquad (9)$$
$$K(\mathbf{p}) = -\tfrac{1}{2}\mathbf{p}^T\mathbf{p}. \qquad (10)$$

When applying the proposed defence to hierarchical models, we update all the latent variables simultaneously. That is, we have $\mathbf{z} = \{\mathbf{z}_1, \ldots, \mathbf{z}_K\}$ and $U(\mathbf{z}) = -\ln p_\theta(\mathbf{x}^a|\mathbf{z}) - \sum_{k=1}^{K} \ln p_\theta(\mathbf{z}_k|\mathbf{z}_{k+1})$.

Eventually, the resulting latents from the HMC are decoded. The steps of the whole process are presented below:

1. (*Defender*) Train a VAE: $q_\phi(\mathbf{z}|\mathbf{x})$, $p(\mathbf{z})$, $p_\theta(\mathbf{x}|\mathbf{z})$.

2. (*Attacker*) For given $\mathbf{x}^r$, calculate the attack $\mathbf{x}^a$ using the criterion equation 3 or equation 4.

3. (*Defender*) Initialize the latent code $\mathbf{z} := \mathbf{z}_0$, where $\mathbf{z}_0 \sim q_\phi(\mathbf{z}|\mathbf{x}^a)$. Then, run $T$ steps of HMC (Algorithm 1) with the step size $\eta$ and $L$ *leapfrog* steps.

The resulting latent code $\mathbf{z}$ can be passed to the decoder to get a reconstruction or to the downstream classification task.

---

**Algorithm 1** One Step of HMC.

---

**Input**: $\mathbf{z}, \eta, L$
  $\mathbf{p} \sim \mathcal{N}(0, I)$.          ▷ Sample the auxiliary variable
  $\mathbf{z}^{(0)} := \mathbf{z}, \mathbf{p}^{(0)} := \mathbf{p}$.
  **for** $l = 1 \ldots, L$ **do**          ▷ Make $L$ steps of *leapfrog*.
    $\mathbf{p}^{(l)} = \mathbf{p}^{(l-1)} - \frac{\eta}{2}\nabla_{\mathbf{z}} U(\mathbf{z}^{(l)})$.
    $\mathbf{z}^{(l)} = \mathbf{z}^{(l)} + \eta\nabla_{\mathbf{p}} K(\mathbf{p}^{(l)})$.
    $\mathbf{p}^{(l)} = \mathbf{p}^{(l)} - \frac{\eta}{2}\nabla_{\mathbf{z}} U(\mathbf{z}^{(l)})$.
  **end for**
                    ▷ Accept new point with prob. $\alpha$.
  $\alpha = \min\left(1, \exp\left(-H(\mathbf{z}^{(L)}, \mathbf{p}^{(L)}) + H(\mathbf{z}^{(0)}, \mathbf{p}^{(0)})\right)\right)$
  $\mathbf{z} = \begin{cases} \mathbf{z}^{(L)} \text{ with probability } \alpha, \\ \mathbf{z}^{(0)} \text{ otherwise.} \end{cases}$
**Return**: $\mathbf{z}$

---

# 4 Experiments

## 4.1 Posterior ratio

We motivate our method by the hypothesis that the adversarial attack "shifts" a latent code to the region of a lower posterior density, while our approach moves it back to a high posterior probability region. In Section 3 we theoretically justify our hypothesis, while here we provide an additional empirical evidence. The true posterior $p(\mathbf{z}|\mathbf{x}^r)$ is not available due to the cumbersome marginal distribution $p(\mathbf{x}^r)$, however, we can calculate the ratio of posteriors because the marginal will cancel out. In our case, we are interested in calculating the posterior ratio between the reference and adversarial latent codes ($\mathbf{z}_1 = \mathbf{z}^r$, $\mathbf{z}_2 = \mathbf{z}^a$) as the baseline, and the posterior ratio between the reference and adversarial

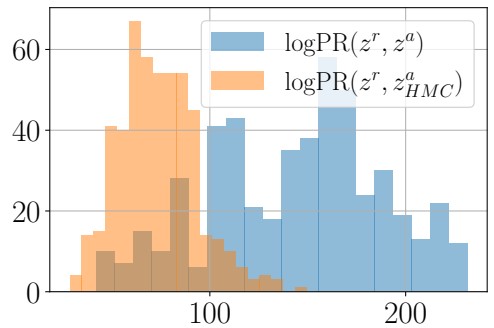

Figure 2: Histograms of the log posterior ratios before HMC (blue) and after HMC (orange) evaluated on the MNIST dataset.

code after applying the HMC ($\mathbf{z}_1 = \mathbf{z}^r$, $\mathbf{z}_2 = \mathbf{z}^a_{\text{HMC}}$). The lower the posterior ratio, the better. For practical reasons, we use the logarithm of the posterior ratio since the logarithm does not change the monotonicity and turns products to sums:

$$\log\text{PR}(\mathbf{z}_1, \mathbf{z}_2) = \log p_\theta(\mathbf{x}^r|\mathbf{z}_1) + \log p(\mathbf{z}_1) - \log p_\theta(\mathbf{x}^r|\mathbf{z}_2) - \log p(\mathbf{z}_2). \tag{11}$$

In Figure 2 we show a plot with two histograms: one with the posterior ratio between the reference and adversarial latent codes ($\mathbf{z}_1 = \mathbf{z}^r$, $\mathbf{z}_2 = \mathbf{z}^a$) in blue, and the second histogram of the posterior ratio between the reference and adversarial code after applying the HMC ($\mathbf{z}_1 = \mathbf{z}^r$, $\mathbf{z}_2 = \mathbf{z}^a_{\text{HMC}}$) in orange. We observe that the histogram has moved to the left after applying the HMC. This indicates that posterior of the adversarial (in the denominator) is increasing when the HMC is used. This is precisely the effect we hoped for and this result provides an empirical evidence in favor of our hypothesis. For more details see C.1.

## 4.2 VAE, $\beta$-VAE and $\beta$-TCVAE

All implementation details and hyperparameters are included in the Appendix D and code repository [2].

---

[2]`https://github.com/AKuzina/defend_vae_mcmc`

**Datasets**  VAEs are trained on the MNIST, Fashion MNIST [44] and Color MNIST datasets. Following [13], we construct the Color MNIST dataset from MNIST by artificially coloring each image with seven colors (all corners of RGB cube except for black).

**Models**  We train vanilla fully convolutional VAEs, as well as $\beta$-VAE [21] and $\beta$-TCVAE [14, 26]. Both $\beta$-VAE and $\beta$-TCVAE modify the ELBO objective to encourage disentanglement. $\beta$-VAE weighs the KL-term in the ELBO with $\beta > 0$. It is said that the larger values of $\beta$ encourage disentangling of latent representations [14] and improve the model robustness as observed by [10]. $\beta$-TCVAE puts a higher weight on the total correlation (TC) term of the ELBO. Penalization of the total correlation was shown to increase the robustness of VAE adversarial attacks [42].

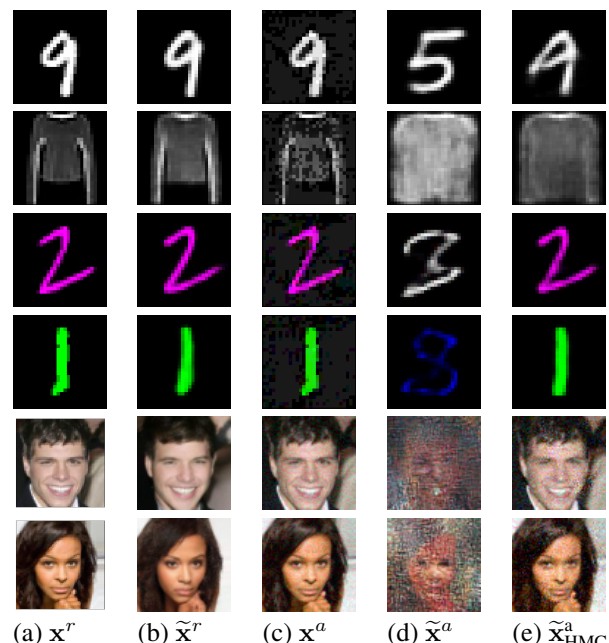

(a) $\mathbf{x}^r$    (b) $\widetilde{\mathbf{x}}^r$    (c) $\mathbf{x}^a$    (d) $\widetilde{\mathbf{x}}^a$    (e) $\widetilde{\mathbf{x}}^a_{\mathrm{HMC}}$

In Appendix D.1 we provide details of the architecture, optimization, and results on the test dataset for VAE trained with different values of $\beta$. We note that the optimal value in terms of the negative log-likelihood (NLL) is always $\beta = 1$. Larger values of $\beta$ are supposed to improve robustness in exchange for the reconstruction quality. When evaluating the robustness of $\beta$-VAE and $\beta$-TCVAE, we train models with $\beta \in \{2, 5, 10\}$. Then, we select the value of $\beta$ that provides the most robust model in terms of the used metric. Next, we apply our defence strategy to this model to observe the potential performance improvement.

Figure 3: Examples of (a) reference points, (b) reconstructions of the reference points, (c) adversarial points, (d) reconstructions of the adversarial points, (e) reconstructions of the adversarial points after the proposed defence (HMC). All the adversarial examples are unsupervised attacks on the encoder. Last two rows contain examples for the NVAE model discussed in Section 4.3

Table 2: Results for unsupervised attack with radius 0.1 and 0.2 on MNIST and Fashion MNIST datasets. We attack the encoder (left) and the downstream classification task (right).
† Our implementation.

|  |  | | $\mathrm{MSSSIM}[\widetilde{\mathbf{x}}^r, \widetilde{\mathbf{x}}^a] \uparrow$ | | ADVERSARIAL ACCURACY $\uparrow$ | | | MSE $\downarrow$ |
|---|---|---|---|---|---|---|---|---|
|  |  | $\|\varepsilon\|$ | 0.1 | 0.2 | 0.0 | 0.1 | 0.2 |  |
| MNIST | | VAE | 0.70 (0.02) | 0.36 (0.03) | **0.90** (0.04) | 0.08 (0.04) | 0.05 (0.03) | 578.7 |
| | | **VAE + HMC** | **0.88** (0.01) | **0.76** (0.02) | 0.76 (0.01) | **0.25** (0.03) | **0.19** (0.03) | **478.1** |
| | | $\beta$-VAE | 0.75 (0.01) | 0.50 (0.03) | 0.90 (0.05) | 0.11 (0.04) | 0.01 (0.01) | 824.2 |
| | | $\beta$-TCVAE† | 0.70 (0.02) | 0.46 (0.03) | 0.86 (0.05) | 0.05 (0.03) | 0.03 (0.02) | 828.4 |
| FASHION MNIST | | VAE | 0.59 (0.03) | 0.47 (0.03) | 0.78 (0.06) | 0.00 (0.01) | 0.01 (0.01) | 814.2 |
| | | **VAE + HMC** | **0.66** (0.03) | **0.54** (0.03) | 0.56 (0.01) | **0.14** (0.02) | **0.13** (0.02) | **764.2** |
| | | $\beta$-VAE | 0.52 (0.03) | 0.41 (0.03) | 0.80 (0.05) | 0.00 (0.01) | 0.00 (0.01) | 1021.1 |
| | | $\beta$-TCVAE† | 0.52 (0.03) | 0.42 (0.03) | **0.84** (0.05) | 0.00 (0.01) | 0.02 (0.02) | 980.4 |

**Attacks on the Encoder**  In the first setup, we assume that the attacker has access to the encoder of the model $q_\phi(z|x)$ [6, 18, 42]. We use the projected gradient descent (PGD) with 50 steps to maximize the symmetric KL-divergence in the unsupervised setting (equation 3). We train 10 adversarial attacks (with different random initialization) for each of 50 reference points. See Appendix D.2 for the details. We report similarity between the reconstruction of the adversarial and reference point as a measure of the robustness (see Section 2.2).

Table 3: Results for unsupervised attack with radius 0.1 and 0.2 on ColorMNIST dataset. We attack the encoder (left) and the downstream classification task (right).
[†] Our implementation.
[*] Values reported in [12], VAE implementation and evaluation protocol may differ.

| | MSSSIM$[\widetilde{x}^r, \widetilde{x}^a]$ ↑ | | ADVERSARIAL ACCURACY ↑ | | | | | | MSE↓ | FID↓ |
| | | | DIGIT | | | COLOR | | | | |
| $\|\varepsilon\|$ | 0.1 | 0.2 | 0.0 | 0.1 | 0.2 | 0.0 | 0.1 | 0.2 | | |
|---|---|---|---|---|---|---|---|---|---|---|
| VAE | 0.36 (0.03) | 0.19 (0.02) | **1.00** (0.00) | 0.04 (0.03) | 0.02 (0.02) | 1.00 (0.00) | 0.06 (0.03) | 0.06 (0.03) | **261** | **2.1** |
| **VAE + HMC** | **0.96** (0.01) | **0.90** (0.01) | 0.42 (0.01) | 0.16 (0.02) | 0.11 (0.02) | 1.00 (0.00) | 0.68 (0.03) | 0.62 (0.03) | **206** | **2.1** |
| $\beta$-VAE | 0.75 (0.01) | 0.5 (0.03) | 0.88 (0.04) | 0.08 (0.04) | 0.05 (0.03) | 1.00 (0.00) | 0.21 (0.06) | 0.18 (0.05) | 366 | 2.4 |
| $\beta$-TCVAE[†] | 0.35 (0.02) | 0.23 (0.02) | 0.94 (0.04) | 0.08 (0.04) | 0.05 (0.03) | 1.00 (0.00) | 0.06 (0.03) | 0.05 (0.02) | 366 | 3.0 |
| $SE_{0.1}$[*] | N/A | N/A | 0.94 (N/A) | 0.89 (N/A) | 0.02 (N/A) | 1.00 (N/A) | 1.00 (N/A) | 0.22 (N/A) | 1372 | 13.0 |
| $SE_{0.2}$[*] | N/A | N/A | 0.95 (N/A) | 0.92 (N/A) | **0.87** (N/A) | 1.00 (N/A) | 1.00 (N/A) | **1.00** (N/A) | 1375 | 11.7 |
| AVAE[*] | N/A | N/A | 0.97 (N/A) | 0.88 (N/A) | 0.55 (N/A) | 1.00 (N/A) | 1.00 (N/A) | 0.88 (N/A) | 1372 | 15.5 |
| $SE_{0.1}$-AVAE[*] | N/A | N/A | 0.97 (N/A) | **0.94** (N/A) | 0.25 (N/A) | 1.00 (N/A) | 1.00 (N/A) | 0.60 (N/A) | 1373 | 13.9 |
| $SE_{0.2}$-AVAE[*] | N/A | N/A | 0.98 (N/A) | **0.94** (N/A) | 0.80 (N/A) | 1.00 (N/A) | 1.00 (N/A) | 0.83 (N/A) | 1374 | 13.9 |
| AVAE-SS[*] | N/A | N/A | 0.94 (N/A) | 0.73 (N/A) | 0.21 (N/A) | 1.00 (N/A) | 1.00 (N/A) | 0.57 (N/A) | 1379 | 12.4 |

**Attacks on the downstream task**  In this setup, we examine how the proposed approach can aleviate the effect of the attack on the downstream tasks in the latent space. For this purpose, we follow the procedure from [12, 13]. Once the VAE is trained, we learn a linear classifier using the mean mappings as features. For the MNIST and Fashion MNIST datasets, we have the 10-class classification problem (digits in the former and pieces of clothing in the latter case). For the ColorMNIST dataset, we consider two classification tasks: the digit classification (10 classes), and the color classification (7 classes). We construct the attack to fool the classifier. See Appendix D.2 for the details.

**Results**  In Tables 2 and 3, we compare our method (VAE + HMC) with the vanilla VAE with other methods in the literature. We report more results and the extended comparison in the Appendix C.2 where we show that our method combined with $\beta$-VAE and $\beta$-TCVAE leads to the increased robustness. For MNIST and Fashion MNIST (Table 2), we observe that the vanilla VAE with the HMC is more robust than $\beta$-VAE and $\beta$-TCVAE. The latter model was shown to be more robust to the latent space attack [42]. Still, in our experiments (on different datasets), we could not observe the consistent improvement over the vanilla VAE, when using it with a single level of latent variables.

In Table 3 we report the result on the ColorMNIST dataset. Here, we additionally compare the adversarial accuracy for our method with the Smooth Encoders (SE) and Autoencoding Variational Autoencoder (AVAE) methods [12, 13]. We notice that these methods provide higher adversarial accuracy. However, we have also observed a large discrepancy in terms of the MSE and FID scores of the model itself compared to our experiments, which we suspect might be a result of a mistake in [12, 13]. [3]

Lastly, we would like to highlight that our defence strategy can be also combined with all the above VAE modifications. One advantage of our approach is that it does not require changing the training procedure of a VAE and, as a result, it does not decrease the quality of the generated images. Moreover, we can apply the same procedure to reconstruct the non-corrupted points and it will improve the reconstruction error. This result can be seen in the Tables 2 and 3 and it goes in line with the results of the [36], where MCMC was used to improve the VAE performance.

### 4.3  Hierarchical VAE: NVAE

**Model and datasets**  In this section, we explore the robustness of the deep hierarchical VAE (NVAE) [39], a specific implementation of a hierarchical VAE that works well for high-dimensional data. We attack models trained on MNIST and CelebA [29] datasets. We use the weights of the pre-trained model provided in the official NVAE implementation[4].

**Attacks construction**  Following [28] we construct adversarial attacks on the hierarchical VAE by considering higher-level latent variables. That being said, we use latent variables

---

[3]We have gotten in touch with the authors, who are kindly working with us to address this issue

[4]The code and model weights were taken from `https://github.com/NVlabs/NVAE`

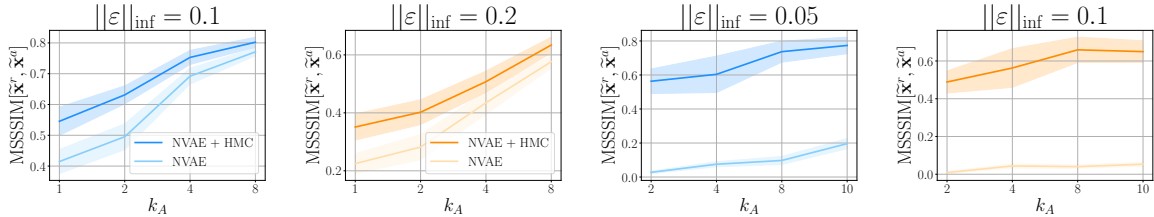

(a) The reconstruction similarity for MNIST          (b) The reconstruction similarity for CelebA

Figure 4: The robustness improvement for the hierarchical model (NVAE) on (a) MNIST and (b) CelebA for two different attack radii. Higher values correspond to more robust representations.

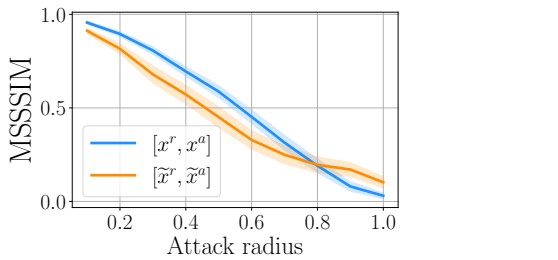 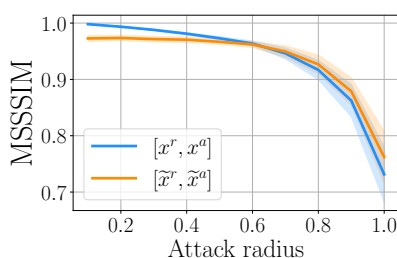

(a) An attacker does not know the defence strategy          (b) An attacker knows the defence strategy

Figure 5: Robustness to adversarial attack (with HMC defence). We report similarity of the reference and adversarial point (blue) and their reconstructions (orange).

$\{\mathbf{z}_{L-k_A}, \mathbf{z}_{L-k_A+1}, \ldots, \mathbf{z}_L\}$ when constructing an attack (equation 3). Otherwise, we follow the same procedure we used for VAEs with a single level of latent variables. We assume that the attacker has access to the model's encoder and uses the symmetric KL-divergence as the objective. The radius of an attack is measured with the $L_{\inf}$ norm. For optimization, we use the projected gradient descent with the number of iterations limited to 50 per point. Further details are reported in the Appendix D.2.

**Results**   In Figure 4, we present reconstruction similarity of the reference and adversarial points for both datasets. We observe that the proposed method consistently improves the robustness of the model to the adversarial attack. This result is in line with our theoretical considerations where a flexible class of variational posteriors could help to counteract adversarial attacks and, eventually, deacrease the bias of the class of models measured in terms of the KL-divergence. Additionally, applying the MCMC can further help us to counteract the attack. In Figure 3 (the two bottom rows), we show how an adversarial point (c) is reconstructed without any defence (d) and with the proposed defence (e). In the depicted samples, we have used the top 10 latent variables ($k_A = 10$) to construct the attack with a radius of 0.05.

### 4.4   Ablation Study: What if the attacker knows the defence strategy?

In our experiments, we rely on the assumption that the attacker does not take into account the defence strategy that we use. We believe that it is reasonable, since the defence requires access to the decoder part of the model, $p_\theta(x|z)$, which is not necessarily available to the attacker.

In this ablation study we verify how the robustness results change if we construct the attack with the access to the defence strategy. We train an adversarial attack with the modified objective 3, which takes into account the HMC step. See Appendix C.3 for more details on the experimental setup.

In Figure 5 we show the experiment results for various attack radii between 0 and 1. We observe that constructing an attack with such an objective is much harder (Figure 5 (b)).

# 5 Discussion

Following the previous works on attacking VAEs [6, 10, 12, 13, 18, 42], we only consider the projected gradient descent as a way to construct the attack. However, more sophisticated adaptive methods [3, 38] were proposed to attack discriminative model and can be potentially applied to VAEs as well. We believe that it is an interesting direction for the future work.

**Objective Function**    For the unsupervised attack on the encoder, we use the symmetric KL-divergence to measure the dissimilarity. However, other options are possible, e.g., the forward or reverse KL-divergence or even $L_2$ distance between the means (see Table 1). In our comparative experiments (see Appendix C.6), we observe that no single objective consistently performs better than others.

**Attack radius**    During the attack construction we seek to obtain a point that will have the most different latent representation or a different predicted class. However, it is also important that the point itself is as similar to the initial reference point as possible. In Appendix C.4, we visualize how the attacks of different radii influence the similarity between the adversarial and reference points. Based on these results, we have chosen the radii which do not allow adversarial points to deviate a lot from the reference (as measured by MSSSIM): $\|\varepsilon\|_{\inf} \leq 0.2$ for the MNIST dataset (which goes in line with the previous works [12, 13]) and $\|\varepsilon\|_{\inf} \leq 0.1$ for the CelebA dataset.

**Number of MCMC steps and Inference Time**    In our approach, we have to select the number of MCMC steps that the defender performs. This parameter potentially can be critical as it influences both the inference time (see Appendix C.7) and the performance (see Appendix C.5). In Figure 6 we show the trade-off between the reconstruction similarity and the inference time. The increase in the inference time is cause by a larger number of HMC steps used (we consider 0, 100, 500 and 1000 steps for this experiment).

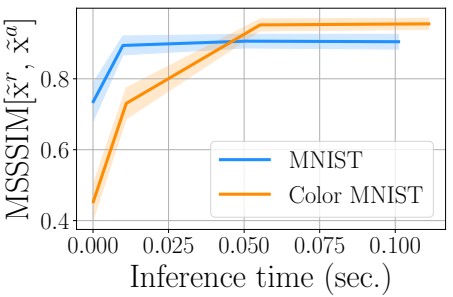

Figure 6: Trade-off between robustness and inference time.

# 6 Conclusion

In this work, we explore the robustness of VAEs to adversarial attacks. We propose a theoretically justified method that allows alleviating the effect of attacks on the latent representations by improving the reconstructions of the adversarial inputs and the downstream tasks accuracy. We experimentally validate our approach on a variety of datasets: both grey-scale (MNIST, Fashion MNIST) and colored (ColorMNIST, CelebA) data. We show that the proposed method improves the robustness of the vanilla VAE models and its various modifications, i.e., $\beta$-VAE, $\beta$-TCVAE and NVAE.

## Acknowledgements

Anna Kuzina is funded by the Hybrid Intelligence Center, a 10-year programme funded by the Dutch Ministry of Education, Culture and Science through the Netherlands Organisation for Scientific Research. Experiments were carried out on the Dutch national e-infrastructure with the support of SURF Cooperative

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
