# A   Theory

We consider an attack, which has an additive structure:

$$\mathbf{x}^a = \mathbf{x}^r + \varepsilon, \tag{12}$$

$$\text{such that } \|\varepsilon\| \le \delta, \tag{13}$$

where $\delta$ is a radius of the attack.

In the vanilla VAE setup we will get the latent code by sampling from $q_\phi(\mathbf{z}|\mathbf{x})$. With our approach, instead, we get a sample from the following distribution:

$$q^{(t)}(\mathbf{z}|\mathbf{x}) = \int Q^{(t)}(\mathbf{z}|\mathbf{z}_0) q_\phi(\mathbf{z}_0|\mathbf{x}) d\mathbf{z}_0, \tag{14}$$

where $Q^{(t)}(\mathbf{z}|\mathbf{z}_0)$ is a transition kernel of MCMC with the target distribution $\pi(\mathbf{z}) = p_\theta(\mathbf{z}|\mathbf{x}) \propto p_\theta(\mathbf{x}|\mathbf{z})p(\mathbf{z})$.

**Lemma 1**   Consider true posterior distributions of the latent code $\mathbf{z}$ for a data point $\mathbf{x}$ and its corrupted version $\mathbf{x}^a$. Assume also that $\ln p_\theta(\mathbf{z}|\mathbf{x})$ is twice differentiable over $\mathbf{x}$ with continuous derivatives at the neighbourhood around $\mathbf{x} = \mathbf{x}^r$. Then the KL-divergence between these two posteriors could be expressed using the small $o$ notation of the radius of the attack, namely:

$$D_{\mathrm{KL}}\left[p_\theta(\mathbf{z}|\mathbf{x}^r)\|p_\theta(\mathbf{z}|\mathbf{x}^a)\right] = o(\|\varepsilon\|). \tag{15}$$

*Proof*
Let us use definition of the KL-divergence:

$$D_{\mathrm{KL}}\left[p_\theta(\mathbf{z}|\mathbf{x}^r)\|p_\theta(\mathbf{z}|\mathbf{x}^a)\right] = \mathbb{E}_{p_\theta(\mathbf{z}|\mathbf{x}^r)} \ln \frac{p_\theta(\mathbf{z}|\mathbf{x}^r)}{p_\theta(\mathbf{z}|\mathbf{x}^a)} \tag{16}$$

Let us introduce $\ln p_\theta(\mathbf{z}|\mathbf{x}) = g(\mathbf{x}, \mathbf{z})$. Assume that this function is differentiable at $\mathbf{x} = \mathbf{x}^r$. Then, we can apply Taylor expansion to $g(\mathbf{x}, \mathbf{z})$ in the point $\mathbf{x}^r$ which yields:

$$g(\mathbf{x}, \mathbf{z}) = g(\mathbf{x}^r, \mathbf{z}) + (\mathbf{x} - \mathbf{x}^r)^T \nabla_\mathbf{x} g(\mathbf{x}, \mathbf{z})\Big|_{\mathbf{x}^r} + R_1(\mathbf{x}, \mathbf{x}^r). \tag{17}$$

The remainder term in the Lagrange form can be written as

$$R_1(\mathbf{x}, \mathbf{x}^r) = \tfrac{1}{2}(\mathbf{x} - \mathbf{x}^r)^T \nabla_{\mathbf{xx}}^2 g(\mathbf{x} + \theta(\mathbf{x} - \mathbf{x}^r), \mathbf{z})(\mathbf{x} - \mathbf{x}^r), \theta \in (0, 1) \tag{18}$$

Under the assumption that $g$ is twice differentiable with the continuous derivatives on the segment around $\mathbf{x} = \mathbf{x}^r$ the remainder term asymptotically converges to zero with $\mathbf{x} \to \mathbf{x}^r$.

$$R_1(\mathbf{x}, \mathbf{x}^r) = o(\|\mathbf{x} - \mathbf{x}^r\|). \tag{19}$$

Then, the log-ratio of two distributions is the following:

$$\ln \frac{p_\theta(\mathbf{z}|\mathbf{x}^r)}{p_\theta(\mathbf{z}|\mathbf{x}^a)} = g(\mathbf{x}^r, \mathbf{z}) - g(\mathbf{x}^a, \mathbf{z}) \tag{20}$$

$$= g(\mathbf{x}^r, \mathbf{z}) - \left(g(\mathbf{x}^r, \mathbf{z}) + (\mathbf{x}^a - \mathbf{x}^r)^T \nabla_\mathbf{x} g(\mathbf{x}, \mathbf{z})\Big|_{\mathbf{x}^r} + o(\|\mathbf{x}^a - \mathbf{x}^r\|)\right). \tag{21}$$

$$= -\varepsilon^T \nabla_\mathbf{x} g(\mathbf{x}, \mathbf{z})\Big|_{\mathbf{x}^r} + o(\|\varepsilon\|). \tag{22}$$

Notice that $\varepsilon^T \nabla_\mathbf{x} g(\mathbf{x}, \mathbf{z})\Big|_{\mathbf{x}^r}$ is the dot product between $\varepsilon$ and $\nabla_\mathbf{x} g(\mathbf{x}, \mathbf{z})\Big|_{\mathbf{x}^r}$, i.e., $\varepsilon^T \nabla_\mathbf{x} g(\mathbf{x}, \mathbf{z})\Big|_{\mathbf{x}^r} = \langle \varepsilon, \nabla_\mathbf{x} g(\mathbf{x}, \mathbf{z})\Big|_{\mathbf{x}^r}\rangle$.

We can now plug this into the KL-divergence definition (16):

$$D_{\mathrm{KL}}\left[p_\theta(\mathbf{z}|\mathbf{x}^r)\|p_\theta(\mathbf{z}|\mathbf{x}^a)\right] = \mathbb{E}_{p_\theta(\mathbf{z}|\mathbf{x}^r)} \left[-\langle \varepsilon, \nabla_\mathbf{x} \ln p_\theta(\mathbf{z}|\mathbf{x})\Big|_{\mathbf{x}^r}\rangle + o(\|\varepsilon\|)\right] \tag{23}$$

$$= -\langle \varepsilon, \underbrace{\mathbb{E}_{p_\theta(\mathbf{z}|\mathbf{x}^r)} \nabla_\mathbf{x} \ln p_\theta(\mathbf{z}|\mathbf{x})\Big|_{\mathbf{x}^r}}_{\mathrm{A}(\mathbf{x}^r)}\rangle + o(\|\varepsilon\|) \tag{24}$$

Note that for (24) to hold we need to make sure that $\mathbb{E}_z R_1 = o(\|\varepsilon\|)$. As follows from (18), this requirement is satisfied if $\mathbb{E}_z \nabla_{\mathbf{xx}}^2 g(\mathbf{x} + \theta(\mathbf{x} - \mathbf{x}^r), \mathbf{z})$ is bounded around $\mathbf{x} = \mathbf{x}^r$.

Let us take a closer to look at the term $\mathrm{A}(\mathbf{x}^r)$ in the equation above:

$$\mathrm{A}(\mathbf{x}^r) = \mathbb{E}_{p_\theta(\mathbf{z}|\mathbf{x}^r)} \nabla_{\mathbf{x}} \ln p_\theta(\mathbf{z}|\mathbf{x})\Big|_{\mathbf{x}^r} \tag{25}$$

$$= \mathbb{E}_{p_\theta(\mathbf{z}|\mathbf{x}^r)} \nabla_{\mathbf{x}} \ln \frac{p_\theta(\mathbf{x}|\mathbf{z})p_\theta(\mathbf{z})}{p_\theta(\mathbf{x})}\Big|_{\mathbf{x}^r} \tag{26}$$

$$= \mathbb{E}_{p_\theta(\mathbf{z}|\mathbf{x}^r)} \nabla_{\mathbf{x}} \ln p_\theta(\mathbf{x}|\mathbf{z})\Big|_{\mathbf{x}^r} - \mathbb{E}_{p_\theta(\mathbf{z}|\mathbf{x}^r)} \nabla_{\mathbf{x}} \ln p_\theta(\mathbf{x})\Big|_{\mathbf{x}^r} \tag{27}$$

$$= \int p_\theta(\mathbf{z}|\mathbf{x}^r) \frac{\nabla_{\mathbf{x}} p_\theta(\mathbf{x}|\mathbf{z})\big|_{\mathbf{x}^r}}{p_\theta(\mathbf{x}^r|\mathbf{z})} d\mathbf{z} - \int p_\theta(\mathbf{z}|\mathbf{x}^r) \frac{\nabla_{\mathbf{x}} p_\theta(\mathbf{x})\big|_{\mathbf{x}^r}}{p_\theta(\mathbf{x}^r)} d\mathbf{z} \tag{28}$$

$$= \int \frac{p_\theta(\mathbf{z})}{p_\theta(\mathbf{x}^r)} \nabla_{\mathbf{x}} p_\theta(\mathbf{x}|\mathbf{z})\Big|_{\mathbf{x}^r} d\mathbf{z} - \frac{\nabla_{\mathbf{x}} p_\theta(\mathbf{x})\big|_{\mathbf{x}^r}}{p_\theta(\mathbf{x}^r)} \underbrace{\int p_\theta(\mathbf{z}|\mathbf{x}^r) d\mathbf{z}}_{=1} \tag{29}$$

$$= \frac{1}{p_\theta(\mathbf{x}^r)} \left[ \int p(\mathbf{z}) \nabla_{\mathbf{x}} p_\theta(\mathbf{x}|\mathbf{z})\Big|_{\mathbf{x}^r} d\mathbf{z} - \nabla_{\mathbf{x}} p_\theta(\mathbf{x})\Big|_{\mathbf{x}^r} \right] \tag{30}$$

$$= \frac{1}{p_\theta(\mathbf{x}^r)} \left[ \mathbb{E}_{p(\mathbf{z})} \nabla_{\mathbf{x}} p_\theta(\mathbf{x}|\mathbf{z})\Big|_{\mathbf{x}^r} - \nabla_{\mathbf{x}} \mathbb{E}_{p(\mathbf{z})} p_\theta(\mathbf{x}|\mathbf{z})\Big|_{\mathbf{x}^r} \right] \tag{31}$$

$$= \frac{1}{p_\theta(\mathbf{x}^r)} \underbrace{\left[ \mathbb{E}_{p(\mathbf{z})} \nabla_{\mathbf{x}} p_\theta(\mathbf{x}|\mathbf{z})\Big|_{\mathbf{x}^r} - \mathbb{E}_{p(\mathbf{z})} \nabla_{\mathbf{x}} p_\theta(\mathbf{x}|\mathbf{z})\Big|_{\mathbf{x}^r} \right]}_{=0} = 0. \tag{32}$$

where we use Bayes rule in 26, 29, log-derivative trick in 28.

We have shown that $\mathrm{A}(\mathbf{x}^r) = 0$, therefore, from equation 24 we have:

$$D_{\mathrm{KL}} \left[ p_\theta(\mathbf{z}|\mathbf{x}^r) \| p_\theta(\mathbf{z}|\mathbf{x}^a) \right] = -\langle \varepsilon, \mathrm{A}(\mathbf{x}^r) \rangle + o(\|\varepsilon\|) = o(\|\varepsilon\|). \tag{33}$$

$\blacksquare$

**Lemma 2** The Total Variation distance (TV) between the variational posterior with MCMC for a given corrupted point $\mathbf{x}^a$, $q^{(t)}(\mathbf{z}|\mathbf{x}^a)$, and the variational posterior for a given data point $\mathbf{x}^r$, $q_\phi(\mathbf{z}|\mathbf{x}^r)$, can be upper bounded by the sum of the following three components:

$$\mathrm{TV} \left[ q^{(t)}(\mathbf{z}|\mathbf{x}^a), q_\phi(\mathbf{z}|\mathbf{x}^r) \right] \leq \mathrm{TV} \left[ q^{(t)}(\mathbf{z}|\mathbf{x}^a), p_\theta(\mathbf{z}|\mathbf{x}^a) \right] \tag{34}$$

$$+ \sqrt{\tfrac{1}{2} D_{\mathrm{KL}} \left[ p_\theta(\mathbf{z}|\mathbf{x}^r) \| p_\theta(\mathbf{z}|\mathbf{x}^a) \right]} \tag{35}$$

$$+ \sqrt{\tfrac{1}{2} D_{\mathrm{KL}} \left[ q_\phi(\mathbf{z}|\mathbf{x}^r) \| p_\theta(\mathbf{z}|\mathbf{x}^r) \right]}. \tag{36}$$

*Proof*
Total variation is a proper distance, thus, the triangular inequality holds for it. For the proof, we apply the triangular inequality twice. First, we use the triangle inequality for $\mathrm{TV} \left[ q^{(t)}(\mathbf{z}|\mathbf{x}^a), q_\phi(\mathbf{z}|\mathbf{x}^r) \right]$, namely:

$$\mathrm{TV} \left[ q^{(t)}(\mathbf{z}|\mathbf{x}^a), q_\phi(\mathbf{z}|\mathbf{x}^r) \right] \leq \mathrm{TV} \left[ q^{(t)}(\mathbf{z}|\mathbf{x}^a), p_\theta(\mathbf{z}|\mathbf{x}^r) \right] + \mathrm{TV} \left[ p_\theta(\mathbf{z}|\mathbf{x}^r), q_\phi(\mathbf{z}|\mathbf{x}^r) \right]. \tag{37}$$

Second, we utilize the triangle inequality for $\mathrm{TV} \left[ q^{(t)}(\mathbf{z}|\mathbf{x}^a), p_\theta(\mathbf{z}|\mathbf{x}^r) \right]$, that is:

$$\mathrm{TV} \left[ q^{(t)}(\mathbf{z}|\mathbf{x}^a), p_\theta(\mathbf{z}|\mathbf{x}^r) \right] \leq \mathrm{TV} \left[ q^{(t)}(\mathbf{z}|\mathbf{x}^a), p_\theta(\mathbf{z}|\mathbf{x}^a) \right] + \mathrm{TV} \left[ p_\theta(\mathbf{z}|\mathbf{x}^a), p_\theta(\mathbf{z}|\mathbf{x}^r) \right]. \tag{38}$$

Combining the two gives us the following upper bound on the initial total variation:

$$\mathrm{TV}\left[q^{(t)}(\mathbf{z}|\mathbf{x}^a), q_\phi(\mathbf{z}|\mathbf{x}^r)\right] \le \mathrm{TV}\left[q^{(t)}(\mathbf{z}|\mathbf{x}^a), p_\theta(\mathbf{z}|\mathbf{x}^a)\right] \tag{39}$$

$$+ \mathrm{TV}\left[p_\theta(\mathbf{z}|\mathbf{x}^a), p_\theta(\mathbf{z}|\mathbf{x}^r)\right] \tag{40}$$

$$+ \mathrm{TV}\left[p_\theta(\mathbf{z}|\mathbf{x}^r), q_\phi(\mathbf{z}|\mathbf{x}^r)\right] \tag{41}$$

Moreover, the Total Variation distance is a lower bound of the KL-divergence (by Pinsker inequality):

$$\mathrm{TV}\left[p(\mathbf{x}), q(\mathbf{x})\right] \le \sqrt{\tfrac{1}{2} D_{\mathrm{KL}}\left[p(\mathbf{x})\|q(\mathbf{x})\right]}. \tag{42}$$

Applying Pinsker inequality to 40 and 41 yields:

$$\mathrm{TV}\left[q^{(t)}(\mathbf{z}|\mathbf{x}^a), q_\phi(\mathbf{z}|\mathbf{x}^r)\right] \le \mathrm{TV}\left[q^{(t)}(\mathbf{z}|\mathbf{x}^a), p_\theta(\mathbf{z}|\mathbf{x}^a)\right] \tag{43}$$

$$+ \sqrt{\tfrac{1}{2} D_{\mathrm{KL}}\left[p_\theta(\mathbf{z}|\mathbf{x}^r)\|p_\theta(\mathbf{z}|\mathbf{x}^a)\right]} \tag{44}$$

$$+ \sqrt{\tfrac{1}{2} D_{\mathrm{KL}}\left[q_\phi(\mathbf{z}|\mathbf{x}^r)\|p_\theta(\mathbf{z}|\mathbf{x}^r)\right]}. \tag{45}$$

∎

**Theorem 1** The upper bound on the total variation distance between samples from MCMC for a given corrupted point $\mathbf{x}^a$, $q^{(t)}(\mathbf{z}|\mathbf{x}^a)$, and the variational posterior for the given real point $\mathbf{x}$, $q_\phi(\mathbf{z}|\mathbf{x})$, is the following:

$$\mathrm{TV}\left[q^{(t)}(\mathbf{z}|\mathbf{x}^a), q_\phi(\mathbf{z}|\mathbf{x}^r)\right] \le \mathrm{TV}\left[q^{(t)}(\mathbf{z}|\mathbf{x}^a), p_\theta(\mathbf{z}|\mathbf{x}^a)\right] + \sqrt{\tfrac{1}{2} D_{\mathrm{KL}}\left[q_\phi(\mathbf{z}|\mathbf{x}^r)\|p_\theta(\mathbf{z}|\mathbf{x}^r)\right]} + o(\sqrt{\|\varepsilon\|}). \tag{46}$$

*Proof*
Combining **Lemma 1** and **2** we get:

$$\mathrm{TV}\left[q^{(t)}(\mathbf{z}|\mathbf{x}^a), q_\phi(\mathbf{z}|\mathbf{x}^r)\right] \underbrace{\le}_{\text{Lemma 2}} \mathrm{TV}\left[q^{(t)}(\mathbf{z}|\mathbf{x}^a), p_\theta(\mathbf{z}|\mathbf{x}^a)\right] \tag{47}$$

$$+ \sqrt{\tfrac{1}{2} D_{\mathrm{KL}}\left[p_\theta(\mathbf{z}|\mathbf{x}^r)\|p_\theta(\mathbf{z}|\mathbf{x}^a)\right]} \tag{48}$$

$$+ \sqrt{\tfrac{1}{2} D_{\mathrm{KL}}\left[q_\phi(\mathbf{z}|\mathbf{x}^r)\|p_\theta(\mathbf{z}|\mathbf{x}^r)\right]} \tag{49}$$

$$\underbrace{=}_{\text{Lemma 1}} \mathrm{TV}\left[q^{(t)}(\mathbf{z}|\mathbf{x}^a), p_\theta(\mathbf{z}|\mathbf{x}^a)\right] \tag{50}$$

$$+ \sqrt{\tfrac{1}{2} o(\|\varepsilon\|)} \tag{51}$$

$$+ \sqrt{\tfrac{1}{2} D_{\mathrm{KL}}\left[q_\phi(\mathbf{z}|\mathbf{x}^r)\|p_\theta(\mathbf{z}|\mathbf{x}^r)\right]}. \tag{52}$$

Note that $\sqrt{\tfrac{1}{2} o(\|\varepsilon\|)} = o(\sqrt{\|\varepsilon\|})$ that gives us the final expression:

$$\mathrm{TV}\left[q^{(t)}(\mathbf{z}|\mathbf{x}^a), q_\phi(\mathbf{z}|\mathbf{x}^r)\right] \le \mathrm{TV}\left[q^{(t)}(\mathbf{z}|\mathbf{x}^a), p_\theta(\mathbf{z}|\mathbf{x}^a)\right] + \sqrt{\tfrac{1}{2} D_{\mathrm{KL}}\left[q_\phi(\mathbf{z}|\mathbf{x}^r)\|p_\theta(\mathbf{z}|\mathbf{x}^r)\right]} + o(\sqrt{\|\varepsilon\|}). \tag{53}$$

∎

# B Background on MCMC

## B.1 Sampling from an unnormalized density with MCMC

Markov Chain Monte Carlo (MCMC) is a class of methods that are used to obtain samples from the density $p(\mathbf{v})$ (also referred to as **target**), which is only known up to a normalizing constant. That is, we have access to $\tilde{p}(\mathbf{v})$, such that $p(\mathbf{v}) = \frac{\tilde{p}(\mathbf{v})}{Z}$ and $Z$ is a typically unknown and hard to estimate normalizing constant. Thus, we construct a Markov Chain with samples $\{\mathbf{v}^{(t)}\}_{t=1}^{T}$ so that they mimic the samples from $p(\mathbf{v})$. To ensure they are proper samples, the stationary distribution of the constructed Markov Chain should be the target distribution $p(\mathbf{v})$.

The most popular way of constructing such Markov Chains is the Metropolis-Hastings (MH) method. The majority of the MCMC methods used in practice can be formulated as a special case of the MH [2]. In the MH method, we introduce a proposal distribution $q(\mathbf{v}^{t+1}|\mathbf{v}^{t})$ to obtain a new sample and then accept it with the following probability:

$$\mathcal{A}(\mathbf{v}^{t}, \mathbf{v}^{t+1}) = \min\{1, \tfrac{p(\mathbf{v}^{t})q(\mathbf{v}^{t+1}|\mathbf{v}^{t})}{p(\mathbf{v}^{t+1})q(\mathbf{v}^{t}|\mathbf{v}^{t+1})}\}. \tag{54}$$

If the point is not accepted, we reuse the previous point, i.e., $\mathbf{v}^{t+1} = \mathbf{v}^{t}$. It can be proven that the resulting chain of correlated samples converges in the distribution to the target density [2].

It is worth mentioning that the performance of the method strongly depends on the choice of the proposal distribution. In higher dimensional spaces, it is especially important to incorporate the information about the geometry of the target distribution into the proposal density to improve the convergence time. The Hamiltonian Monte-Carlo (HMC) [31] is known to be one of the most efficient MCMC methods. It uses gradient of a target distribution in the proposal to incorporate the information about the geometry of the space.

The idea of the HMC is to introduce an auxiliary variable $\mathbf{p}$ with a known density (usually assumed to be the standard Gaussian) and the joint distribution formulated as follows:

$$p(\mathbf{v}, \mathbf{p}) = \frac{1}{Z} \exp(-U(\mathbf{v})) \exp(-K(\mathbf{p})), \tag{55}$$

with:

$$K(\mathbf{p}) = -\frac{1}{2}\mathbf{p}^{T}\mathbf{p}, \tag{56}$$

$$U(\mathbf{v}) = -\log \tilde{p}(\mathbf{v}). \tag{57}$$

We obtain samples $(\mathbf{v}, \mathbf{p})$ using the Hamiltonian dynamics [31] that describes how the $\mathbf{v}$ and $\mathbf{p}$ change over time for the given Hamiltonian $H(\mathbf{v}, \mathbf{p}) = U(\mathbf{v}) + K(\mathbf{p})$, namely:

$$\dot{\mathbf{v}} = \frac{\partial H}{\partial \mathbf{p}}, \tag{58}$$

$$\dot{\mathbf{p}} = -\frac{\partial H}{\partial \mathbf{v}}. \tag{59}$$

For the practical implementation, these continuous-time equations are approximated by discretizing the time using $L$ small steps of size $\eta$. The discretization method that is often used is called the *leapfrog*.

## B.2 The MCMC and Variational Autoencoders

In this paper, we use the MCMC to sample from the posterior distribution $p_{\theta}(\mathbf{z}|\mathbf{x}^{a})$. That is, in our case $\mathbf{v} = \mathbf{z}$ and $\tilde{p}(\mathbf{v}) = p_{\theta}(\mathbf{x}^{a}|\mathbf{z})p(\mathbf{z})$. The HMC is a widely applied method to sampling from an unknown posterior distribution in deep learning (e.g. [24]). A lot of effort was already done in combining variational inference with MCMC (and more specifically with HMC). Hamiltonian Variational Inference [36, 43] was proposed in order to obtain a more flexible variational approximation.

Different approaches were proposed to use HMC during VAE training. [23] approximate the gradients of the likelihood and avoid the use of variational approximation. [11] propose an unbiased estimate for the ELBO gradient, which allows training Hamiltonian Variational Autoencoder. [35] propose an alternative objective, which uses a contrastive divergence instead of standard KL-divergence.

In this work we are not changing the training procedure, instead, we propose to only use HMC during evaluation.

**A possible extension to discrete latent spaces**   Some VAEs operate on discrete latent spaces, a very popular example would be a VQ-VAE [40]. However, the classical HMC that we use in our experiments is not able to sample from a discrete distribution. Therefore, other MCMC methods should be used in this case, such as population-based MCMC [4], modifications of HMC[32] or Langevin Dynamics [45].

### B.3   Mode optimization

In this work we hypothesise that adversarial attacks move latent codes to the region of low probability and we use HMC to get a sample from the high posterior probability region. However, another strategy can be to find the posterior mode instead. Here we explain, what was our motivation to not use this approach.

**Posterior modes similarity**   Ideally, we would like to obtain a sample from the variational posterior $q_\phi(\mathbf{z}|\mathbf{x}^r)$, because our decoder was trained to produce reconstructions from such latent codes. At the same time, VAE was trained to match this variational posterior to the true one $p_\theta(\mathbf{z}|\mathbf{x}^r)$. However, both these distributions are not available to us, since we observe attacked point $\mathbf{x}^a$ instead of the reference $\mathbf{x}^r$.

Instead, we sample from $p_\theta(\mathbf{z}|\mathbf{x}^a)$ and show theoretically that the resulting samples are close (in terms of total variation distance) to the "goal" ones. However, that does guarantee that their modes are the same. Therefore, obtaining the mode of $p_\theta(\mathbf{z}|\mathbf{x}^a)$ is not necessarily a mode of $q_\phi(\mathbf{z}|\mathbf{x}^r)$. Thus, the fact that the HMC allows us to "wander" around that mode may be beneficial.

**Concentration of measure**   During reconstruction, we get a sample from $q(\mathbf{z}|\mathbf{x})$ and pass it to the decoder, thus, a mode can actually be a bad latent code for these purposes. Instead, ideally, we want to get a sample from the typical set where most of the probability mass is concentrated. In theory, the HMC allows us to do that.

**Randomness**   The HMC adds a source of randomness to our defence strategy that potentially makes it harder to attack. This is supported by our experiment in Section 4.4

# C Additional results

## C.1 Posterior ratio

We motivate our method by the hypothesis that the adversarial attack "shifts" a latent code to the region of a lower posterior density, while our approach moves it back to a high posterior probability region. In Section 3 we theoretically justify our hypothesis, while here we provide an additional empirical evidence.

In order to verify our claim that applying an MCMC method allows us to counteract attacks by moving a latent code from a region of a lower posterior probability mass to a region of a higher density, we propose to quantify this effect by measuring the ratio of posteriors for $\mathbf{z}_1$ and $\mathbf{z}_2$. The true posterior $p(\mathbf{z}|\mathbf{x}^r)$ is not available due to the cumbersome marginal distribution $p(\mathbf{x}^r)$, however, we can calculate the ratio of posteriors because the marginal will cancel out, namely:

$$\mathrm{PR}(\mathbf{z}_1, \mathbf{z}_2) = \frac{p_\theta(\mathbf{z}_1|\mathbf{x}^r)}{p_\theta(\mathbf{z}_2|\mathbf{x}^r)} \tag{60}$$

$$= \frac{p_\theta(\mathbf{x}^r|\mathbf{z}_1)p(\mathbf{z}_1)}{p_\theta(\mathbf{x}^r|\mathbf{z}_2)p(\mathbf{z}_2)}. \tag{61}$$

In our case, we are interested in calculating the posterior ratio between the reference and adversarial latent codes ($\mathbf{z}_1 = \mathbf{z}^r$, $\mathbf{z}_2 = \mathbf{z}^a$) as the baseline, and the posterior ratio between the reference and adversarial code after applying the HMC ($\mathbf{z}_1 = \mathbf{z}^r$ , $\mathbf{z}_2 = \mathbf{z}^a_{\mathrm{HMC}}$). The lower the posterior ratio, the better. For practical reasons, we use the logarithm of the posterior ratio (the logarithm does not change the monotonicity and turns products to sums):

$$\log \mathrm{PR}(\mathbf{z}_1, \mathbf{z}_2) = \log p_\theta(\mathbf{x}^r|\mathbf{z}_1) + \log p(\mathbf{z}_1) - \log p_\theta(\mathbf{x}^r|\mathbf{z}_2) - \log p(\mathbf{z}_2). \tag{62}$$

We present results on the log-posterior-ratio calculated on the MNIST dataset. In Figure 7 we show a plot with two histograms: one with the posterior ratio between the reference and adversarial latent codes ($\mathbf{z}_1 = \mathbf{z}^r$, $\mathbf{z}_2 = \mathbf{z}^a$) in blue, and the second histogram of the posterior ratio between the reference and adversarial code after applying the HMC ($\mathbf{z}_1 = \mathbf{z}^r$ , $\mathbf{z}_2 = \mathbf{z}^a_{\mathrm{HMC}}$) in orange.

We observe that the histogram has moved to the left after applying the HMC. This indicates that posterior of the adversarial (in the denominator) is increasing when the HMC is used. This is precisely the effect we hoped for and this result provides an empirical evidence in favor of our hypothesis.

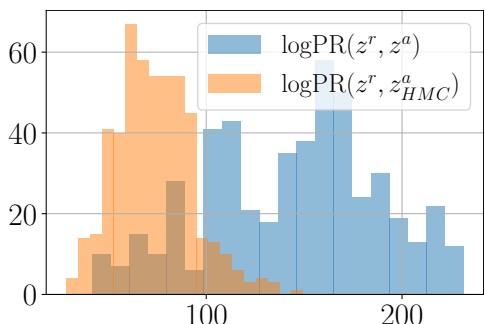

Figure 7: Histograms of the log posterior ratios without HMC (blue) and with HMC (orange) evaluated on MNIST dataset.

**Experimental Details**   For this experiment we construct 500 adversarial attacks with the radius 0.1 on the encoder of the VAE trained on MNIST dataset. We run 500 HMC steps with the same hyperparameters as mentioned in Table 8 to obtain $\mathbf{z}^a_{HMC}$.

**Statistical Analisys**   We performed a two-sample Kolmagorov-Smirnov test with the null hypothesis that two histograms are drawn from the same distribution. As an alternative hypothesis is that the underlying distributions are different. Choosing the confidence level of 95% results in the rejection of the null hypothesis (p-value is equal to 0.029) in favour of the alternative: two histograms were not drawn from the same distribution.

## C.2  Detailed results for $\beta$-VAE and $\beta$-TCVAE

In this section we report extended results for MNIST, FashionMNIST and ColorMNIST datasets. We train VAE, $\beta$-VAE and $\beta$-TCVAE on three datasets: MNIST, FashionMNIST and ColorMNIST. Then, we compare the robustness to adversarial attack with and without HMC. We present all the result with the standard error in Table 4. On Figures 8, 9 we show visually how HMC improve the robustness for each dataset and model.

Table 4: Robustness results on MNIST, Fashion MNIST and Color MNIST datasets. We perform unsupervised attack with radius 0.1 (top) and 0.2 (bottom). We attack the encoder (left) and the downstream classification task (right). Higher values correspond to more robust models.

| | | MSSSIM$[\tilde{\mathbf{x}}^r, \tilde{\mathbf{x}}^a]\uparrow$ | | | ADVERSARIAL ACCURACY $\uparrow$ | | COLOR MNIST | |
| | | MNIST | FASHION MNIST | COLOR MNIST | MNIST | FASHION MNIST | DIGIT | COLOR |
|---|---|---|---|---|---|---|---|---|
| $\|\epsilon\|_{\inf}=0.1$ | VAE | 0.70 (0.02) | 0.59 (0.03) | 0.36 (0.03) | 0.08 (0.04) | 0.00 (0.01) | 0.04 (0.03) | 0.06 (0.03) |
| | + HMC | **0.88** (0.01) | **0.66** (0.03) | **0.96** (0.01) | 0.25 (0.03) | **0.14** (0.02) | 0.16 (0.02) | 0.68 (0.03) |
| | $\beta$-VAE | 0.75 (0.01) | 0.52 (0.03) | 0.50 (0.04) | 0.11 (0.04) | 0.00 (0.02) | 0.08 (0.04) | 0.21 (0.06) |
| | + HMC | 0.84 (0.01) | 0.64 (0.03) | 0.92 (0.03) | **0.30** (0.03) | 0.13 (0.02) | 0.14 (0.02) | 0.66 (0.04) |
| | $\beta$-TCVAE | 0.70 (0.02) | 0.52 (0.03) | 0.35 (0.02) | 0.05 (0.03) | 0.01 (0.01) | 0.08 (0.04) | 0.06 (0.03) |
| | + HMC | 0.79 (0.02) | **0.66** (0.03) | **0.96** (0.01) | 0.25 (0.04) | 0.13 (0.02) | **0.22** (0.03) | **0.81** (0.02) |
| $\|\epsilon\|_{\inf}=0.2$ | VAE | 0.36 (0.03) | 0.47 (0.03) | 0.19 (0.02) | 0.05 (0.03) | 0.01 (0.01) | 0.02 (0.02) | 0.06 (0.03) |
| | + HMC | **0.76** (0.02) | **0.54** (0.03) | **0.90** (0.01) | **0.19** (0.03) | **0.13** (0.02) | 0.11 (0.02) | 0.62 (0.03) |
| | $\beta$-VAE | 0.50 (0.03) | 0.41 (0.03) | 0.38 (0.04) | 0.01 (0.01) | 0.00 (0.01) | 0.05 (0.03) | 0.18 (0.05) |
| | + HMC | 0.69 (0.03) | 0.50 (0.03) | 0.87 (0.01) | 0.16 (0.03) | 0.12 (0.02) | **0.15** (0.02) | 0.56 (0.04) |
| | $\beta$-TCVAE | 0.45 (0.03) | 0.42 (0.03) | 0.20 (0.02) | 0.03 (0.02) | 0.02 (0.02) | 0.05 (0.03) | 0.05 (0.03) |
| | + HMC | 0.65 (0.03) | 0.52 (0.03) | 0.87 (0.01) | 0.16 (0.04) | 0.11 (0.02) | 0.14 (0.02) | **0.72** (0.03) |

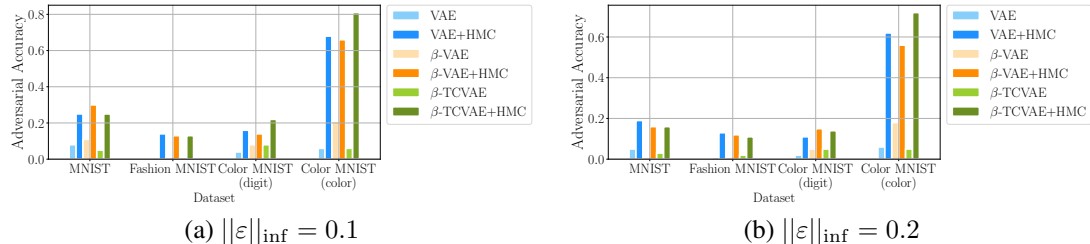

(a) $\|\varepsilon\|_{\inf}=0.1$      (b) $\|\varepsilon\|_{\inf}=0.2$

Figure 8: Improvement of the Reconstruction Similarity after the proposed defence. We fix the attack radius to be equal to (a) 0.1 and (b) 0.2. Higher values correspond to a more robust representations.

(a) $\|\varepsilon\|_{\inf}=0.1$      (b) $\|\varepsilon\|_{\inf}=0.2$

Figure 9: Improvement of the Adversarial Accuracy after proposed defence. We fix the attack radius to be equal to (a) 0.1 and (b) 0.2.

## C.3 What if the attacker knows the defence strategy?

In our experiments we relied on the assumption that attack does not take into account the defence strategy that we use. We believe that it is reasonable, since defence requires access to the decoder part of the model ($p_\theta(x|z)$), which is not necessarily available to the attacker.

However, one may assume that the defence strategy is known to the attacker. In this case, it is reasonable to verify whether the robustness results change. In the conducted experiment we show that it is vastly more complicated to attack the encoder with taking the MCMC defence into account. We train the unsupervised attack (3). The attack has access to the encoder and MCMC defence:

$$f(x) = q^{(t)}(\mathbf{z}|\mathbf{x}) = \int Q^{(t)}(\mathbf{z}|\mathbf{z}_0)q_\phi(\mathbf{z}_0|\mathbf{x})d\mathbf{z}_0, \tag{63}$$

where $Q^{(t)}(\mathbf{z}|\mathbf{z}_0)$ is MCMC kernel.

Then, given the attack radius $\delta$, we train the attack using the following objective:

$$\varepsilon^* = \arg \max_{\|\varepsilon\|_{\inf} < \delta} \|\widetilde{z}^a - \widetilde{z}^r\|^2, \tag{64}$$

$$\widetilde{z}^a \sim q^{(t)}(\mathbf{z}|\mathbf{x}^r + \varepsilon), \tag{65}$$

$$\widetilde{z}^r \sim q^{(t)}(\mathbf{z}|\mathbf{x}^r). \tag{66}$$

The similarity results of these attacks are plotted in Figure 11. We observe that the reconstructed reference and adversarial points have approximately the same similarity (measured by MSSSIM) as the initial points $\mathbf{x}^r$ and $\mathbf{x}^a$, which indicates that the attacks were unsuccessful.

However, if we use the same objective, but omit the MCMC step (e.g $t = 0$ in eq. 65 and 66), then, as observed in Figure 10, the attack becomes much more successful (Figure 10 (a)), but we can fix it with the proposed defence (Figure 10 (b)).

It is interesting to compare how the attacked points look in both cases, especially as we increase the radius of the attack. In Figure 12, we plot attack on two reference points for radius values in $\{0.1, 0.6, 0.8, 1.0\}$. When the attacker does not use MCMC (left), it just learns to add more and more noise to the image, which eventually makes it meaningless.

When we use MCMC during an attack, the situation is different. The adversarial input is almost indistinguishable from the reference point for a small radius. After each gradient update, the attacker runs a new MCMC, which moves point closer to the region of high posterior probability, but may follow a different trajectory every time. Eventually, it makes it harder to learn an additive perturbation $\varepsilon$. However, as we increase the attack radius, we observe a very interesting effect. Instead of meaningless noise, the attacker learns to change the digit. For instance, we see how $4$ is transformed into $0$ in the first example and into $9$ in the second. This way, the attacker ensures that the MCMC will move the latent far away from the reference latent code, which now has a different posterior distribution.

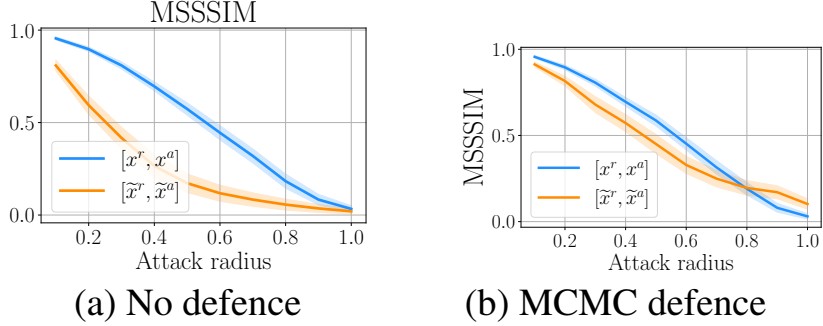

(a) No defence  (b) MCMC defence

Figure 10: Adversarial attack, if attacker **does not use MCMC**. We report similarity of the reference and adversarial point before forward pass (blue) and after forward pass (orange).

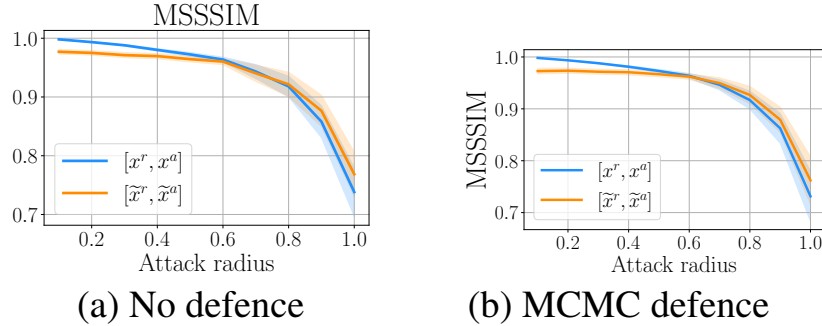

Figure 11: Adversarial attack, if attacker **uses MCMC**. We report similarity of the reference and adversarial point before forward pass (blue) and after forward pass (orange).

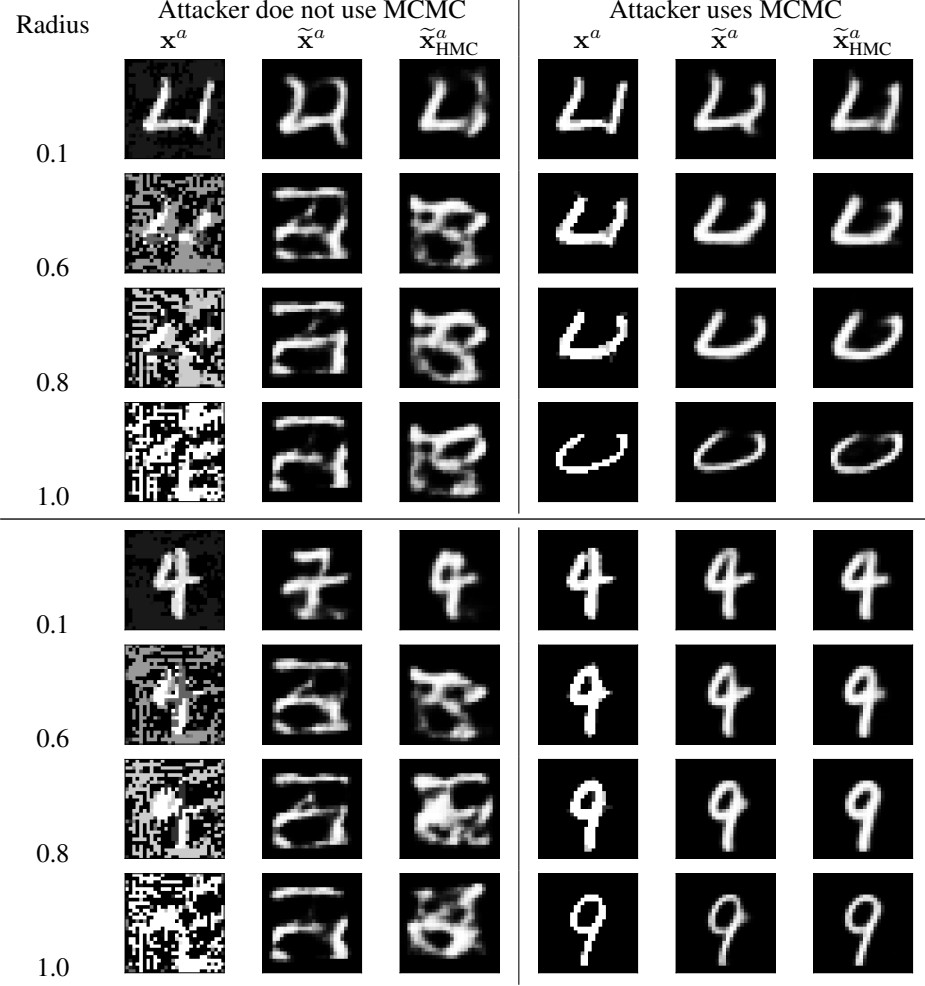

Figure 12: Examples of adversarial point and their reconstructions, when attacker does not use MCMC (left) and when attacker uses MCMC(right).

## C.4 Which attack radius should be considered?

In out experiments, we use attacks with the radius 0.1 and 0.2 for all the models except for CelebA dataset, where radii 0.05 and 0.1 were considered. Here, we provide additional experiment to justify this choice. In Figure 13 (a) we show the similarity between the reference point and the adversarial point. We observe that for CelebA the similarity drops faster than for the MNIST. Further, if we look at the example plotted in Figure 14, we can clearly notice that with the radius 0.2 CelebA image is already containing a lot of noise. At the same time, we observe (Figure 13 (b)) that reconstruction similarity, which indicates the success of the attack, drops relatively fast when the radius of the attack increases.

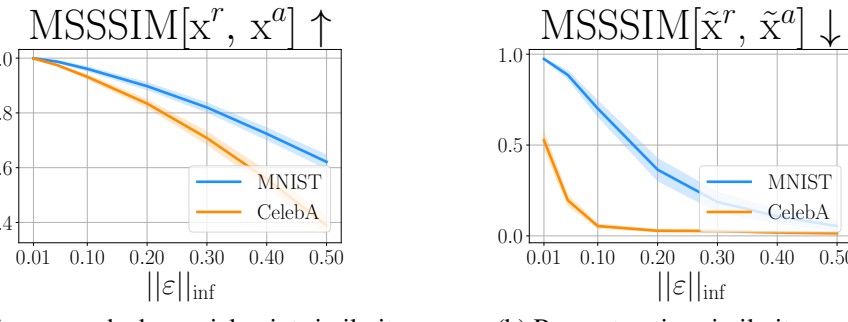

(a) Reference and adversarial point similarity      (b) Reconstruction similarity

Figure 13: Average images similarity (a) before it is passed to VAE and (b) after image is encoded and decoded back. We consider unsupervised attack on the encoder with the radiuses ranging from 0.01 to 0.5 for MNIST and CelebA datasets.

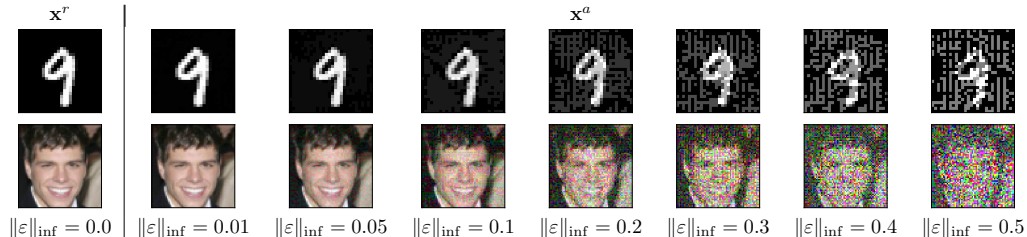

Figure 14

## C.5 How many HMC steps are required for a defence?

One of the main hyperparameters of the proposed approach is number of steps of MCMC that the defender does. We have conducted experiments with MNIST and Color MNSIT dataset to see how the robustness metrics change when we increase number of HMC steps from 0 to 200. As we can see from the Figure 15, there is always a considerable jump between 0 steps (no defence) and 100 steps (lowest number of steps considered). However, as we continue making steps, we do not observe further improvement of the metrics.

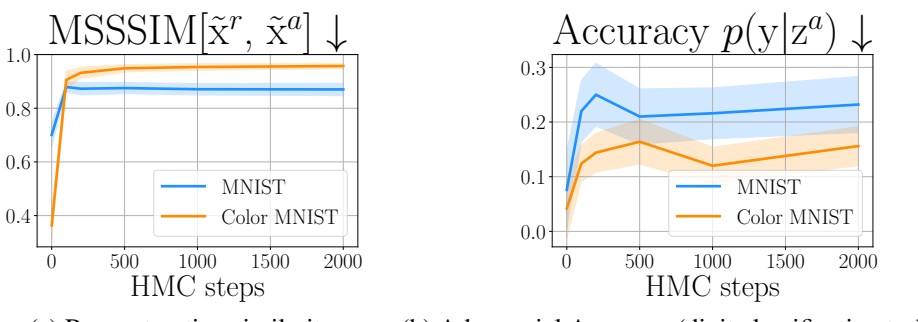

(a) Reconstruction similarity.  (b) Adversarial Accuracy (digit classification task).

Figure 15: Example of the reference point (leftmost column) and adversarial points for different raduises of the attack.

### C.6 Comparison of objective functions

This section compares different objective functions that can be used to construct adversarial attacks on VAE. In general, in both supervised and unsupervised setting, we need to measure the difference between variational posterior in the adversarial point $q(\mathbf{z}|\mathbf{x}^a)$ and a point from the dataset (either a target or reference point). We consider a Gaussian encoder, and the simplest way to compare two Gaussian distributions is to measure the distance between their means. To take into account the variances, we can use the KL-divergence. It is a non-symmetric metric. Thus, we have two options: to use the forward or reverse KL. Finally, it is also possible to consider the symmetrical KL divergence that is an average between the two.

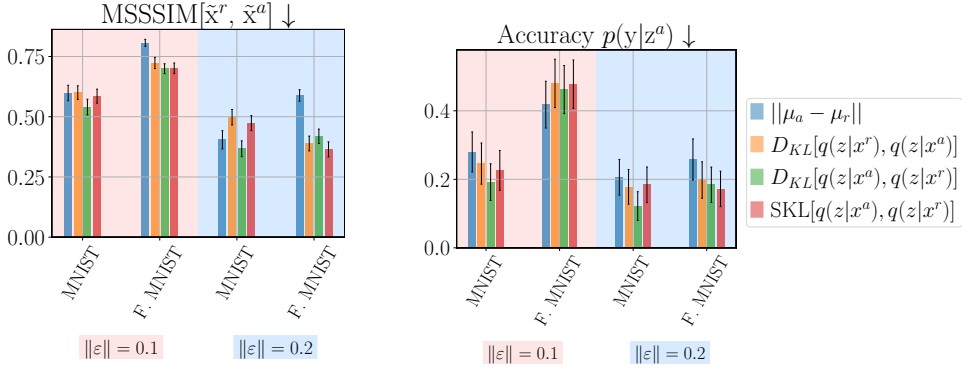

(a) Reconstructions similarity: adversarial point and the reference.

(b) Adversarial Accuracy.

Figure 16: Comparison of different objectives function used to train an attack. Arrows represent the direction of the successful attack.

In Figure 16, we measure how successful the attacks are in terms of the proposed metrics. We use arrows in the plot titles to indicate desirable values of the metric for a successful attack. We compare supervised and unsupervised attacks on VAE trained on MNIST and fashion MNIST datasets. We observe that there is no single objective function that consistently outperforms others.

### C.7 Inference Time

Even though our approach does not require changing the training procedure, it has influence on the inference time. In practice, this can be a limiting factor. Therefore, in Table 5 we report the computational overhead during the inference time. We measure the inference time in seconds per test point without HMC ($T = 0$) and for different budgets ($T = \{100, 500, 1000\}$).

Table 5: Inference wall-clock time of the VAE for various number of MCMC steps ($T$).

| $T$ | 0 | 100 | 500 | 1000 |
|---|---|---|---|---|
| | VAE | | | |
| MNIST | 0.0001 | 0.0099 | 0.0505 | 0.1011 |
| COLOR MNIST | 0.0001 | 0.0110 | 0.0553 | 0.1111 |
| | NVAE | | | |
| CELEBA | 0.429 | 6.512 | 31.551 | 63.031 |

# D  Details of the experiments

## D.1  Training VAE models

**Architecture**  We use the same fully convolutional architecture with latent dimension 64 for MNIST, FashionMNISt and ColorMNIST datasets. In Table 6, we provide detailed scheme of the architecture. We use `Conv(3x3, 1->32)` to denote convolution with kernel size 3x3, 1 input channel and 32 output channels. We denote stride of the convolution with `s`, padding with `p` and dilation with `d`. The same notation applied for the transposed convolutions (`ConvTranspose`). ColorMNIST has 3 input channels, so the first convolutional layer in the encoder and the last of the decoder are slightly different. In this cases values for ColorMNIST are report in parenthesis with the red color.

Table 6: Convolutional architecture for VAE trained on MNIST, Fashion MNIST and ColorMNIST datasets.

| Encoder | Decoder |
|---|---|
| `Conv(3x3, 1(3)->32, s=2, p=1)` | `ConvTranspose(3x3,64->128,s=1,p=0, d=2)` |
| `ReLU()` | `ReLU()` |
| `Conv(3x3, 32->64, s=2, p=1)` | `ConvTranspose(3x3,128->96,s=1,p=0)` |
| `ReLU()` | `ReLU()` |
| `Conv(3x3, 64->96, s=2, p=1)` | `ConvTranspose(3x3,96->64,s=1,p=1)` |
| `ReLU()` | `ReLU()` |
| `Conv(3x3,96->128,s=2,p=1)` | `ConvTranspose(4x4,64->32,s=2,p=1)` |
| `ReLU()` | `ReLU()` |
| $\mu_z \leftarrow$ `Conv(3x3,128->64,s=2,p=1)` | `ConvTranspose(4x4,31->1(3),s=2,p=1)` |
| $\log \sigma_z^2 \leftarrow$ `Conv(3x3,128->64,s=2,p=1)` | $\mu_x \leftarrow$ `Sigmoid() (Identity())` |

**Optimization**  We use Adam to perform the optimization. We start from the learning rate $5e-4$ and reduce it by the factor of 2 if the validation loss does not decrease for 10 epochs. We train a model for 300 epochs with the batch size 128. In Table 7, we report the values of the test metrics for VAEs trained on MNIST, Fashion MNIST and Color MNIST.

For calculating the FID score, we use `torchmetrics` library: `https://torchmetrics.readthedocs.io/en/latest/references/modules.html#frechetinceptiondistance`.

Table 7: Test performance of the $\beta$-VAE and $\beta$-TCVAE with different values of $\beta$. Negative loglikelihood is estimated with importance sampling ($k = 1000$) as suggested in [9].

| | $\beta$ | MNIST $-\log p(\mathbf{x})$ | MSE | FASHION MNIST $-\log p(\mathbf{x})$ | MSE | COLOR MNIST $-\log p(\mathbf{x})$ | MSE | FID |
|---|---|---|---|---|---|---|---|---|
| | 1 | **88.3** | 578.6 | **232.8** | 814.3 | **54.87** | 261.3 | 2.09 |
| $\beta$-VAE | 2 | 89.3 | 824.2 | 234.1 | 1021.1 | 55.6 | 365.6 | 2.4 |
| | 5 | 100.6 | 1485.1 | 241.8 | 1457.8 | 63.6 | 586.1 | 2.5 |
| | 10 | 126.8 | 2498.9 | 248.7 | 1842.3 | 88.7 | 936.2 | 2.4 |
| $\beta$-TCVAE | 2 | 89.3 | 828.4 | 233.6 | 980.4 | 55.8 | 366.4 | 3.0 |
| | 5 | 96.7 | 1325.4 | 238.2 | 1024.6 | 63.0 | 574.8 | 2.0 |
| | 10 | 107.2 | 1686.1 | 247.5 | 1570.0 | 76.5 | 806.2 | 2.2 |

## D.2 Adversarial Attacks and Defence Hyperparameters

In Table 8, we report all the hyperparameter values that were used to attack and defend VAE models.

In all the experiments we randomly select reference points from the test dataset. We also ensure that the resulting samples are properly stratified — include an even number of points from each of the classes. For each reference point, we train 10 adversarial inputs with the same objective function but different initialization.

We use projected gradient descent to learn the adversarial attacks. Optimization parameters were the same for all the datasets and models. They are presented in Table 8.

We choose HMC to defend the model against the trained attack. We perform $T$ steps of HMC with the step size $\eta$ and $L$ leapfrog steps. Where indicated, we adapt the step size after each step of HMC using the following formula:

$$\eta_t = \eta_{t-1} + 0.01 \cdot \frac{\alpha_{t-1} - 0.9}{0.9} \cdot \eta_{t-1}, \tag{67}$$

where $\alpha_t$ is the acceptance rate at step $t$. This way we increase the step size if the acceptance rate is higher than 90% and decrease it otherwise. When adaptive steps size is used, a value in the table indicates the $\eta_0$.

Table 8: Full list of hyperparameters for attack construction and the defence.

|  |  | VAE | | | NVAE | |
|---|---|---|---|---|---|---|
|  |  | MNIST | Fashion MNIST | Color MNIST | MNIST | CelebA |
|  | # of reference points | 50 | 50 | 50 | 50 | 20 |
|  | # of adversarial points | 500 | 500 | 500 | 500 | 200 |
|  | Radius norm ($\|\cdot\|_p$) | inf | inf | inf | inf | inf |
|  | Radius | $\{0.1, 0.2\}$ | $\{0.1, 0.2\}$ | $\{0.1, 0.2\}$ | $\{0.1, 0.2\}$ | $\{0.05, 0.1\}$ |
| Optimization (PGD) | Optimizer | | | SGD | | |
|  | Num. steps | | | 50 | | |
|  | $\varepsilon$ initialization | | | $\mathcal{N}(0, 0.2 \cdot I)$ | | |
|  | Learning rate (lr) | | | 1 | | |
| Defence (HMC) | Num. steps ($T$) | 500 | 1000 | 1000 | 2000 | 1000 |
|  | Step size $\eta$ | 0.1 | 0.05 | 0.05 | 1e-4 | 1e-4 |
|  | Num. Leapfrog steps ($L$) | 20 | 20 | 20 | 20 | 1 |
|  | Adaptive step size | True | True | True | True | False |