# OpenReview forum: "Alleviating Adversarial Attacks on Variational Autoencoders with MCMC"
_NeurIPS.cc/2022/Conference — NeurIPS 2022 Accept_

### Official Review · Reviewer_bo5D · 2022-07-05

**Rating:** 6
**Confidence:** 4
**Soundness:** 3 good
**Presentation:** 3 good
**Contribution:** 2 fair

**Summary:**

The paper proposes to defend against adversarial attacks on variational autoencoders by doing Hamiltonian Monte Carlo in the latent space to move the attacked sample back to a more high probability region. The authors compare the effects across different datasets and attack methodologies and show that it improves robustness. Additionally, the authors produce theoretical results that show that doing HMC should help against attacks.

**Questions:**

If the logic is that samples from low density latent spaces are worse why is HMC used which just aims to produce independent samples. Wouldn't instead an optimisation technique that tries to maximize the posterior be more helpful? HMC might even reduce the posterior of the sample.

I find the results in Section 4.3. very surprising. How does knowing the defence make the attacker so much weaker? I would expect the opposite. Do the authors have any insight on this?

Small comments:
- Since MSSSIM and the Adv. accuracy have arrows pointing up it would be good to add arrows pointing down for MSE and FID in Table 2 and 3.
- It would be better to have MSSSIM as the y-axis label than the title in Figure 4.

**Limitations:**

The authors emphasize that their method only changes the inference and not the training compared to other defenses. While this has obvious advantages the paper should also discuss the downsides, e.g. that in practical applications often you would trade longer training time if it avoids a computationally demanding MCMC step for every inference call.

**Strengths And Weaknesses:**

**Strengths**

The paper explains the method well and compares it with several other methods in detail.
The theory part gives a good explanation as to why the method can lead to improved robustness.
A method that only changes the inference part has the huge advantage that it can be retrofitted to trained models which can be very helpful when it is hard to retrain a model or in cases where the model can't be changed.
Using MCMC to improve robustness is an interesting direction that can be very helpful for the community if further explored.

**Weaknesses**

The paper claims multiple times that because training isn't changed the performance for non-attacked inputs won't be decreased. I don't understand that logic, as the inference defence would have to be applied to non-attacked inputs as well, as usually you won't know beforehand if the image has been attacked or not. In lines 145-146 even claim that it could improve results for non-attacked images. Looking at all the results it seems like this is never tested. Figure 1 doesn't apply MCMC to the non-attacked input, and the MSE and FID values in Table 2 and 3 are exactly the same for VAE and VAE + HMC which seems unlikely if MCMC was used. If the results are really not decreased for non-attacked samples then this should be shown.
While the method increases robustness, the robust accuracy is very low in some cases compared to other methods e.g. just 0.16 or 0.11 for ColorMNIST accuracy in Table 3 compared to other methods getting values around over 0.9 or 0.8 respectively. While the authors show that these methods result in high MSE and FID errors this might not be relevant in some applications. It would be interesting to include nominal accuracy in the results to show if the increased robustness leads to lower nominal accuracy for these methods. This would also again be interesting to see if the HMC steps have an effect on nominal accuracy.

---

> ### Author Response · Authors · 2022-08-02
> **Answer to Reviewer bo5D**
>
> Dear reviewer bo5D,
>
> First of all, we would like to thank you very much for your review. We are excited that you find this direction interesting and potentially helpful for the community. We sincerely appreciate the time you spent evaluating our work and very much appreciate your comments.
>
> Below we answer your questions, comments and concerns. We also upload the updated version of the manuscript, where we mark all mentioned changes in blue.
>
> **[Weaknesses]**
>
> **In lines 145-146 even claim that it could improve results for non-attacked images. Looking at all the results it seems like this is never tested. Figure 1 doesn't apply MCMC to the non-attacked input, and the MSE and FID values in Table 2 and 3 are exactly the same for VAE and VAE + HMC which seems unlikely if MCMC was used. If the results are really not decreased for non-attacked samples then this should be shown.**
>
> We agree that the results on non-attacked images were not presented. Indeed, MSE and FID scores for VAE and VAE + HMC were both calculated without running the HMC.
>
> *MSE*. We’ve properly evaluated VAE + HMC on the test set and updated the numbers in Tables 2 and 3 accordingly. We *do* observe that test error got lower. This is expected because the MCMC generates samples from $\propto p_{\theta}(x|z)p(z)$ and for the non-attacked points from the test set this results in higher likelihood values (on average). This result also goes in line with [1], where MCMC was used to improve the performance of the VAE.
>
> [1] T. Salimans, D. Kingma, and M. Welling. Markov chain monte carlo and variational inference: Bridging the gap. In International conference on machine learning, pages 1218–1226. PMLR, 2015
>
> *FID*. We would like to note that the FID score measures how good the samples from the model (not the reconstructions) are. We do not run the HMC during sampling, since there is no input $x$ in that case. In this case, $z$ were sampled from the prior and then reconstructed.
>
> *Figure 1*.
> Figure 1 was mainly added to illustrate intuitively how the proposed method works.
>
>
> **It would be interesting to include nominal accuracy in the results to show if the increased robustness leads to lower nominal accuracy for these methods. This would also again be interesting to see if the HMC steps have an effect on nominal accuracy.**
>
> Could you perhaps clarify what you mean by “nominal accuracy”, please?
> In the experiments, we report the adversarial accuracy (see Tables 2 and 3). In the column with the radius equal to 0, we report how a method performs without any adversarial attacks. We do observe that HMC lowers the accuracy with radius 0. We believe that this is due to the fact that the classifier was trained on the *mean* values of the variational posterior (to be comparable with the previous works) and not on the samples from $q$.
>
> **[Questions]**
>
> **If the logic is that samples from low density latent spaces are worse why is HMC used which just aims to produce independent samples. Wouldn't instead an optimisation technique that tries to maximize the posterior be more helpful? HMC might even reduce the posterior of the sample.**
>
> This is a very fair point and an interesting discussion. We do not have a rigorous answer why the HMC is definitely better, but intuitively we have the following motivation to use it instead of just maximizing the posterior (which is an interesting idea though):
>
> 1. *[Posterior modes similarity]*
>
> Ideally, we would like to obtain a sample from the variational posterior $q(z|x^r)$, because our decoder was trained to produce reconstructions from such latent codes. At the same time, VAE was trained to match this variational posterior to the true one $p_{\theta}(z|x^r)$. However, both these distributions are not available to us, since we observe attacked point $x^a$ instead of the reference $x^r$.
>
> Instead, we sample from $p_{\theta}(z|x^a)$ and show theoretically that the resulting samples are close (in terms of total variation distance) to the "goal" ones. However, that does guarantee that their modes are the same. Therefore, obtaining the mode of $p_{\theta}(z|x^a)$  is not necessarily a mode of $q(z|x^r)$. Thus, the fact that the HMC allows us to “wander” around that mode may be beneficial.
>
> 2. *[Concentration of measure]*
>
> During reconstruction, we get a sample from $q(z|x)$ and pass it to the decoder, thus, a mode can actually be a bad latent code for these purposes. Instead, ideally, we want to get a sample from the typical set where most of the probability mass is concentrated. In theory, the HMC allows us to do that.
>
>
> 3. *[Randomness]*
>
> The HMC adds a source of randomness to our defence strategy that potentially makes it harder to attack. This is supported by our experiment in Section 4.3 (see the next question for the discussion).

---

> > ### Author Response · Authors · 2022-08-02
> > **Answer to Reviewer bo5D (2)**
> >
> > **I find the results in Section 4.3. very surprising. How does knowing the defence make the attacker so much weaker? I would expect the opposite. Do the authors have any insight on this?**
> >
> > We believe that this is due to the fact that it is harder to attack with additional randomness introduced by the HMC. In our setup, an attacker uses projected gradient descent (PGD) maximizing the $L_2$ distance of two samples obtained by running the HMC in the latent space. After each PGD iteration, the $x^a$ point is updated and, thus, the attacker has to run a new Markov chain. Therefore, randomness induced by this procedure makes PGD very ineffective if not impossible to use for attacking purposes.
> >
> > **Small comments:**
> > * **Since MSSSIM and the Adv. accuracy have arrows pointing up it would be good to add arrows pointing down for MSE and FID in Table 2 and 3.**
> > * **It would be better to have MSSSIM as the y-axis label than the title in Figure 4.**
> >
> > Thank you for pointing that out, we’ve updated the manuscript accordingly.
> >
> > **[Limitaitons]**
> >
> > **…method only changes the inference and not the training compared to other defenses. While this has obvious advantages the paper should also discuss the downsides, e.g. that in practical applications often you would trade longer training time if it avoids a computationally demanding MCMC step for every inference call.**
> >
> >
> > We agree that we did not properly discuss this. We’ve added the validations times to Appendix C.7 and referenced it in the discussion section. The page limit does not allow us to include it in the main body of the paper, but we will gladly do it in camera-ready version.

---

> > > ### Comment · Reviewer_bo5D · 2022-08-05
> > > **Answer**
> > >
> > > Thank you for the rebuttal and the changes to the paper. I have upgraded my score and would be willing to set it to a 7 if some limitations are made clear in the main paper (see below).
> > >
> > > **HMC on non-attacked images**
> > > I thank the authors for changing the results and adding the reference about MCMC improving VAE performance.
> > >
> > > **Nominal accuracy**
> > > Thank you for clarifying, I did miss the attack radius going to 0. The reduction in accuracy is a bit worrying but I guess that is a common trade-off when using defences like this. I think this needs to be made clear in the main paper though.
> > >
> > > **Using HMC instead of optimizations**
> > > I think this needs to be made clear in the paper, as it mentions multiple times that attacks move the latents to lower probability regions which would make an optimization method the natural choice for me.
> > >
> > > **Increased inference time**
> > > I think this is an important limitation that should be discussed, even by just in a single sentence, in the main paper. Increasing the inference time several fold could make this method unusable in many practical settings.

---

> > > > ### Author Response · Authors · 2022-08-08
> > > > **Discussion**
> > > >
> > > > Dear reviewer,
> > > >
> > > > We are really happy that you appreciated our rebuttal. We've updated the paper taking into account additional comments.
> > > >
> > > >
> > > > **Using HMC instead of optimisation**
> > > >
> > > > We’ve added this discussion to Appendix B.3 and shortly to the beginning of section 3.
> > > >
> > > > **Increased inference time**
> > > >
> > > > We have made several adjustments regarding the inference time discussion:
> > > > - We have added a Figure to the discussion (section 5), where we plot reconstruction similarity with the inference time on the x-axis
> > > > - We have added inference time for the NVAE model to Appendix C.7
> > > >
> > > > We did not add the same plot for adversarial accuracy vs inference time due to the page limit. If the paper is accepted we will add the second plot to this section in the camera-ready version.

---

### Official Review · Reviewer_knXo · 2022-07-09

**Rating:** 7
**Confidence:** 3
**Soundness:** 4 excellent
**Presentation:** 3 good
**Contribution:** 3 good

**Summary:**

The paper presents the hypothesis that adversarial attacks on VAEs push the latent code into low probability areas. According to this hypothesis, the attack can be mitigated by pushing back the latent code into more probable region of the latent space. To do so, MCMC is applied in inference time. The results demonstrate that the images generated by the decoder from the "corrected" code are indeed more accurate as reconstructions.

**Questions:**

What is the sensitivity of the method to the percentage of corrected examples?

**Limitations:**

Yes.

**Strengths And Weaknesses:**

The paper propose a simple hypothesis, which opens an avenue into a simple mitigation technique. The paper is clearly written, and the approach seems to be theoretically justified and empirically convincing. The proposed method has the advantage of being applied post-hoc on a pretrained model without having to modify the encoder or re-train it.

It would be beneficial to more directly assess the original hypothesis, i.e., to what extent the latent codes produced by the model after the attack are less probable? In addition, it would be beneficial to consider alternative correction methods, e.g. performing a k-means clustering on the latent space and pushing each point towards the center of each assigned cluster.

---

> ### Author Response · Authors · 2022-08-02
> **Answer to Reviewer knXo**
>
> Dear reviewer knXo,
>
> First of all, we would like to thank you very much for your review. We sincerely appreciate the time you spent evaluating our work and very much appreciate your comments.
>
> Below we provide answers to your questions, comments and concerns.  We also upload the updated version of the manuscript, where we mark all mentioned changes in blue.
>
> **It would be beneficial to more directly assess the original hypothesis, i.e., to what extent the latent codes produced by the model after the attack are less probable?**
>
> Thank you for the great suggestion. Since the true posterior is not available to estimate we consider the posterior ratio: $PR(z^r, z^a) = \tfrac{p(z^r | x^r)}{p(z^a | x^r)}$. Plotting the histogram (Figure 5 in Appendix C.1, blue), we observe that this ratio is always larger than 1, which supports our hypothesis that the adversarial latent code lies in a less probable posterior region.  We also show that after running HMC the ratio decreases, indicating the latent code has moved to a more probable posterior region.
>
> Due to the current page limit, we cannot add this experiment to the main body of the paper. However, we will eagerly do it in the camera ready version.
>
> **In addition, it would be beneficial to consider alternative correction methods, e.g. performing a k-means clustering on the latent space and pushing each point towards the center of each assigned cluster.**
>
> Thank you for this idea. It sounds interesting, however, it would require some clarification on how it should work exactly (e.g., how to pick k, what data k-means should be trained on). Without knowing the specifics, it is hard to say more about the idea, however, we are open to further discussion.
>
> **What is the sensitivity of the method to the percentage of corrected examples?**
>
> In general, the method treats each point separately, therefore, the ratio of the corrupted/real examples should not influence its performance. In the provided experiments all the points were corrupted since the main goal was to assess how the robustness to attacks changes with the proposed approach.
>
> In addition, the MSE column in Tables 2 and 3 reports the reconstruction error for the whole test set. We observe that applying HMC reduces the reconstruction error of the model.

---

> > ### Comment · Reviewer_knXo · 2022-08-09
> > **Response**
> >
> > Thank you for the clarifications. I keep my original positive assessment of the paper.

---

### Official Review · Reviewer_yegK · 2022-07-11

**Rating:** 6
**Confidence:** 4
**Soundness:** 4 excellent
**Presentation:** 3 good
**Contribution:** 3 good

**Summary:**

This paper proposes a defense against supervised and unsupervised adversarial attacks in VAEs – in the former a classifier trained on the latent space of the encoder is attacked, and in the later the latent codes themselves are attacked such that the decoded image looks different. The defense is motivated by the hypothesis that adversarial codes are located at low probability regions of the posterior distribution’s support, and consequently an effective defense is to move any code to a high probability region. The paper achieves this by using a particular MCMC approach – Hamiltonian Monte Carlo – as a post-processing sampling step on the codes generated by the encoder of a VAE. The theoretical results support the validity of the approach, and the empirical results show that it is effective in reducing the success of an adversary in both supervised and unsupervised attacks across several datasets.

**Questions:**

1. Is Figure 1 an illustration or an actual experiment? If it's an experiment, the “we observe...” part in the caption needs some more evidence, such as reporting posterior probability ratio assigned to reference codes over perturbed codes. This will also directly support your hypothesis.

2. In Algorithm 1, inside the for loop, $z^l$ should have superindex l-1 in the right side of the first and second line, and $p^l$ should be a temporary variable except in the left side of the last line (since it is overwriting itself)? Also, it is important to report the suggested or default values (used in your experiments) for the hyperparameters inside Algorithm 1's definition.

3. Theorem 3 should be Theorem 1?

4. What are the time, computation and memory costs of the proposed defense? How do they scale with the architecture (particularly in case of hierarchical VAEs)?

5. Can you comment on the applicability of the defense on VQVAE, or conditional VAEs with discrete latents?


**Limitations:**

Limitations are sufficiently discussed, except for the time and computation costs mentioned above.

**Strengths And Weaknesses:**

The paper's proposed defense is simple and effective, and the theoretical analysis sufficiently backs up the defense. To my knowledge, this particular defense is novel, and it can be very significant given that it is independent of the VAE architecture and causes minor degradations in the generative quality. The paper also takes care to study the effect of several hyperparameters and design choices.

A main concern for me is the lack of a discussion on the inference time. The proposed defense will inevitably be slower than baseline VAEs due to the sequential sampling process, which typically takes 1000 steps per Table 7 in Appendix C.2, and requires at least 100 steps per Figure 12 in Appendix B.4. The analysis presented in Tables 2 and 3 would be much more substantial if the inference time is included. I also think that the choice of the particular MCMC algorithm – HMC – needs to be discussed in more detail versus other sampling strategies, given its central role in the proposed defense.

---

> ### Author Response · Authors · 2022-08-02
> **Answer to Reviewer yegK**
>
> Dear reviewer yegK,
>
> First of all, we would like to thank you very much for your review. We are excited that you find our approach novel and potentially significant to the field. We sincerely appreciate the time you spent evaluating our work and very much appreciate your comments.
>
> We answer your questions, comments and concerns below. We also upload the updated version of the manuscript, where we mark all mentioned changes in blue.
>
> **[Weaknesses]**
>
> **A main concern for me is the lack of a discussion on the inference time. … The analysis presented in Tables 2 and 3 would be much more substantial if the inference time is included.**
>
> This is, indeed, the main limitation of the method, which we did not discuss in enough detail. We’ve updated the manuscript, including the inference times for two datasets in Appendix C.7 and mention it in the discussion section. The current page limit does not allow us to add this experiment to the main body of the paper. However, we will eagerly do that in the camera-ready version.
>
> **I also think that the choice of the particular MCMC algorithm – HMC – needs to be discussed in more detail versus other sampling strategies, given its central role in the proposed defense.**
>
> We would like to highlight that our main idea and its theoretical justification do not depend on the specific choice of an MCMC method. We propose to sample from the unnormalized posterior $p_{\theta}(z|x^a)$ and show that this is a reasonable thing to do both theoretically and empirically.
> The HMC [1]  is known to be one of the most efficient MCMC methods. One of its advantages is that it uses gradient information [1] to explore the space more efficiently. Furthermore, it is widely applicable in deep learning, including combination with the VAE and Variational inference [2], [3]. We’ve added a short discussion on this topic to the updated version of the paper (Appendix B.1.)
>
> [1] Neal, Radford M. "MCMC using Hamiltonian dynamics." Handbook of markov chain monte carlo 2.11 (2011): 2.
>
> [2] Salimans, Tim, Diederik Kingma, and Max Welling. "Markov chain monte carlo and variational inference: Bridging the gap." International conference on machine learning. PMLR, 2015.
>
> [3] Ruiz, Francisco, and Michalis Titsias. "A contrastive divergence for combining variational inference and mcmc." International Conference on Machine Learning. PMLR, 2019.
>
> **[Questions]**
>
> **1. Is Figure 1 an illustration or an actual experiment? If it's an experiment, the “we observe...” part in the caption needs some more evidence, such as reporting posterior probability ratio assigned to reference codes over perturbed codes. This will also directly support your hypothesis.**
>
> Figure 1 is an actual experiment for which we train a VAE with 2D latent space. But the purpose of this experiment was to add an intuitive illustration of how the method works. Evaluating posterior probability ratios for two adversarial attacks presented on the Figure 1 gives us $65$ and $31$ before the HMC and $-22$ and $2$ after the HMC. To be precise, we’ve calculated $\log \tfrac{p_{\theta}(z^r|x^r)}{p_{\theta}(z^a|x^r)}$ and $\log \tfrac{p_{\theta}(z^r|x^r)}{p_{\theta}(z^a_{HMC}|x^r)}$. The decrease in the posterior ratio supports our hypothesis that after MCMC latent code moves to a more probable posterior density. However, we are worried that adding these results to the introduction will make this part too cluttered.
> Since the proposition of calculating the posterior probability ration is very interesting and insightful, we decided to make a proper experiment in which we evaluate the posterior probability ratio for the “normal sized” VAE and plot histograms of the resulting numbers (see appendix C.1 in the updated manuscript). The obtained results provide empirical evidence in favor of our hypothesis.
> The page limit does not allow us to include it in the main body of the paper, but we will gladly do it in camera-ready version.
>
> Please, let us know what you think of this. Maybe there exists a concise way to depict posterior ratios on the Figure 1 in a concise way.
>
> **2. In Algorithm 1, inside the for loop, $z^l$ should have superindex $l-1$ in the right side of the first and second line, and $p^l$ should be a temporary variable except in the left side of the last line (since it is overwriting itself)?**
>
> Yes, that was a typo. We’ve fixed this in the updated version of the manuscript.
>
> **Also, it is important to report the suggested or default values (used in your experiments) for the hyperparameters inside Algorithm 1's definition.**
>
> We provide the values of the hyperparameters in Appendix D.2 (see Table 8).
>
> **3. Theorem 3 should be Theorem 1?**
>
> You are right. It was automatically generated by LaTex. The issue is fixed now.

---

> > ### Author Response · Authors · 2022-08-02
> > **Answer to Reviewer yegK  (2)**
> >
> >
> > **4. What are the time, computation and memory costs of the proposed defense? How do they scale with the architecture (particularly in case of hierarchical VAEs)?**
> >
> > As mentioned above, the inference time increases with the number of steps. Furthermore, each step of the HMC requires computing the gradient of the decoder w.r.t. the latent code ($\nabla_z p_{\theta}(x^a|z)$) as shown in Section 3 of the paper. So the complexity of the method increases with the size of the latent space, but it still requires less computation budget than the backpropagation step during training as the number of parameters of the decoder is much larger.
> > We have updated the discussion section mentioning this limitation and included inference times for our method compared to pure VAE in Appendix C.7 of the paper.
> >
> > **5. Can you comment on the applicability of the defense on VQVAE, or conditional VAEs with discrete latents?**
> >
> > Conceptually, the idea of sampling from the posterior distribution $p_{\theta}(z|x^a) \propto p_{\theta}(z^a|z)p(z)$ to alleviate the effect of the adversarial attack is not limited to continuos latent spaces.
> > However, the classical HMC that we use in our experiments is not able to sample from the discrete distribution. Therefore, other MCMC methods should be used in this case. Examples of such MCMC methods are:
> >
> > [1] Nishimura, Akihiko, David B. Dunson, and Jianfeng Lu. "Discontinuous Hamiltonian Monte Carlo for discrete parameters and discontinuous likelihoods." Biometrika 107.2 (2020): 365-380.
> >
> > [2] Zhang, Ruqi, Xingchao Liu, and Qiang Liu. "A Langevin-like Sampler for Discrete Distributions." International Conference on Machine Learning. PMLR, 2022.
> >
> > In general, we find the problem of sampling from discrete latent space a very interesting research question on its own. We are also not aware of the works which show how robust VAEs with discrete latents (such as VQVAE) are to adversarial attacks. This is an excellent future research direction.

---

> > > ### Comment · Reviewer_yegK · 2022-08-08
> > > **Response**
> > >
> > > Thank you for adding the extra content, I strongly suggest that the discussion of HMC and inference time vs. adversarial accuracy be included in the main body.
> > >
> > > Table 5 of C.7 is very helpful in clarifying the time requirements, combining it with Figure 13 of C.5 and Table 2, 3 would be even more informative (for example one could plot adversarial accuracy vs. time for several VAE architectures and for several methods of adversarial defense, this would clearly illustrate the advantages/disadvantages of the different defenses).
> > >
> > > Appendix B.1 and B.2 are nice and essential additions, it is a good idea to include a brief version of them in the main body.
> > >
> > > Figure 5 of C.1 looks very good. I suggest adding details of the experiment (number of samples, HMC steps, etc.) as well as some statistical significance analysis.
> > >
> > > I don’t understand this statement: “but it still requires less computation budget than the backpropagation step during training as the number of parameters of the decoder is much larger.”
> > >
> > > Finally, I tend to keep my current score on two accounts: first, lack of fair time comparison with baselines, specifically Table 2 and 3 should give a measure of inference time; second, not including more recent VAEs (NVAE and VQVAE) in the analysis – and lack of scale – makes the potential impact of this paper limited in my opinion. Nonetheless, I think this is a good paper with solid contributions, hence the score of 6.

---

> > > > ### Author Response · Authors · 2022-08-08
> > > > **Dicussion**
> > > >
> > > > Dear reviewer,
> > > >
> > > > Thank you for appreciating our rebuttal.
> > > >
> > > > **Inference time**
> > > >
> > > > Thank you for the great suggestion to combine inference time and performance from Figure 13. We have made several adjustments in the paper:
> > > >
> > > > - We have added a Figure to the discussion (section 5), where we plot reconstruction similarity with the inference time on the x-axis
> > > > - We have added inference time for the NVAE model to Appendix C.7
> > > >
> > > > We did not add the same plot for adversarial accuracy vs inference time due to the page limit. If the paper is accepted we will add the second plot to this section in the camera-ready version.
> > > >
> > > >
> > > > **Figure 5 in C.1**
> > > >
> > > > For this plot, we used the same setup as in the first column in Table 2 (attacks on the encoder with radius 0.1, 50 reference points with 10 attacks on each of them and 500 HMC steps).  We’ve updated the C.1 with all the experimental details for clarity.
> > > >
> > > > We’ve performed a two-sample Kolmagorov-Smirnov test with the null hypothesis that two histograms are drawn from the same distribution. An alternative hypothesis is that the two distributions are not identical. Choosing the confidence level of 95% results in the rejection of the null hypothesis (p-value is equal to 0.029) in favour of the alternative: two histograms were not drawn from the same distribution.
> > > >
> > > > The appendix is updated correspondingly.

---

### Official Review · Reviewer_d2M4 · 2022-07-11

**Rating:** 5
**Confidence:** 3
**Soundness:** 2 fair
**Presentation:** 2 fair
**Contribution:** 3 good

**Summary:**

This paper proposes Hamiltonian Monte Carlo (HMC) strategy for defending pre-trained encoder of VAE from adversarial attacks.
The authors provides theoretical analysis to substantiate the proposed method, which alleviating the adversarial attacks.
Also, the authors conduct experiments on various benchmark datasets to show the robustness of the proposed defending strategy.

**Questions:**

- What is $t$ in $q^t$, the number of MCMC steps?
- In the argument in Line 172-173, true posterior $p_\theta (z|x^a)$ is unknown. How can one tell $q^t (z|x^a) \rightarrow p_\theta (z|x^a)$ as $t \rightarrow \infty$, if $t$ implies the number of MCMC steps? I might miss something, but since Theorem 3 takes huge part of this paper, hence I respectfully ask the authors to address the question with theoretical backups.
- The $\beta$-VAEs are not designed for defending the adversarial attacks, rather they are for the disentangling purpose. I'm not sure if comparing VAE+HMC against $\beta$-VAEs is a fair comparison.
- Can the same strategy be applied to VAEs with other priors with explicit distributions? For example, priors of discrete probabilities such as Dirichlet distribution or Softmax computation after Gaussian sampling. Or, exponential families in general.
- What is the input scale? 0-1 for every experiment?
- In the caption of Table 2: radius 0.1 **(top)** and 0.2 **(bottom)**, perhaps typos?
- What is AVAE-SS in Table 3?
- How does the performance change when apply the proposed HMC method to the other VAE variants in Table 3?
- Typo in Line 332, a dash -.


**Limitations:**

The authors discussed the limitation of their work in Section 5, and it seems there are no potential negative societal impact on their work.

**Strengths And Weaknesses:**

- The work studies on interesting and significant topic, and the approach seems very simple to apply. I do believe that this kind of work gives positive social impact, for example, attacks like vicious data generation.
- The result also seems promising, including the one in Section 4.3. However, it would be much better if the authors also provide the qualitative result, the one in Figure 2, on high-resolution images regarding the experiment on NVAE.
- HMC seems unfamiliar in recent deep learning community, and it would be better to add the gentle explanation about HMC in the background section.
- My concern (and also questions) on the paper is pointed at Question #2 and #3, which I might have not fully understood. Please read below questions.
- The paper is generally well-written, but some details are missing which are the keys to understand the entire paper. See Question section below.

---

> ### Author Response · Authors · 2022-08-02
> **Answer to Reviewer d2M4**
>
> Dear reviewer d2M4,
>
> First of all, we would like to thank you very much for your review. We are excited that you find our results promising and the topic that we study significant. We sincerely appreciate the time you spent evaluating our work and very much appreciate your comments.
>
> Please find answers to your questions, comments and concerns below. We also upload the updated version of the manuscript, where we mark all mentioned changes in blue.
>
> **[Strengths and weaknesses]**
>
> **…would be much better if the authors also provide the qualitative result, the one in Figure 2, on high-resolution images regarding the experiment on NVAE.**
>
> The last two rows of Figure 2 contain examples of the NVAE model. We’ve combined all qualitative examples in a single figure to save space.  We mention it in Section 4.2, however, the caption of the figure does not explain that, so we updated it accordingly.
>
> **HMC seems unfamiliar in recent deep learning community, and it would be better to add the gentle explanation about HMC in the background section.**
>
> We have added as many details on the HMC as required for our method in Section 3. However, as we can see now, some parts are still unclear, so we have added a short background section on that topic to the Appendix B.1.
>
>
> We would like to highlight that our method can be used with other MCMC methods as well. We use the HMC [1] since it is known to be one of the most efficient MCMC methods. One of its advantages is that it uses gradient information [1] to explore the space more efficiently. Furthermore, it was also applied in deep learning (e.g. see [2] as a good example of using it for SOTA neural networks), including combination with the VAE and Variational inference [3], [4].
>
> [1] Neal, Radford M. "MCMC using Hamiltonian dynamics." Handbook of markov chain monte carlo 2.11 (2011)
>
> [2] Izmailov, Pavel, et al. "What are Bayesian neural network posteriors really like?." International conference on machine learning. PMLR, 2021.
>
> [3] Salimans, Tim, Diederik Kingma, and Max Welling. "Markov chain monte carlo and variational inference: Bridging the gap." International conference on machine learning. PMLR, 2015.
>
> [4] Ruiz, Francisco, and Michalis Titsias. "A contrastive divergence for combining variational inference and mcmc." International Conference on Machine Learning. PMLR, 2019.
>
> **[Questions]**
> **What is $t$ in $q^t$,the number of MCMC steps?**
>
> Yes, it is the number of steps. We have updated the manuscript mentioning this (see lines 148, 163)
>
> **In the argument in Line 172-173, true posterior $p_{\theta}(z|x^a)$ is unknown. How can one tell $q^t(z|x^a) \rightarrow p_{\theta}(z|x^a)$ as $t \rightarrow \inf$, if $t$ implies the number of MCMC steps? I might miss something, but since Theorem 3 takes huge part of this paper, hence I respectfully ask the authors to address the question with theoretical backups.**
>
> This follows from the properties of the MCMC. The aim is to sample from the unnormalized density; in our case we sample from $p_{\theta}(z|x^a) \propto p_{\theta}(x^a|z)p(z))$. When we run a Markov chain for $t$ steps, we get a sample from the distribution that we call $q^t$. This is not exactly the distribution we want to sample from, but it can be proved (see for example [1]) that it converges to the target distribution as $t \rightarrow \inf$.
> We added a detailed description of this idea to Appendix B.1 of the paper.
>
> [1] Andrieu, Christophe, et al. "An introduction to MCMC for machine learning." Machine learning 50.1 (2003): 5-43
>
> **The β-VAEs are not designed for defending the adversarial attacks, rather they are for the disentangling purpose. I'm not sure if comparing VAE+HMC against β-VAEs is a fair comparison.**
>
> We agree that $\beta$-VAE was designed for disentangling purposes. However, prior works on adversarial attacks have shown that disentanglement improves the robustness. This is the reason we include this comparison. We’ve also performed experiments where we combine  $\beta$-VAE with our proposed approach (see Table 4 in appendix C.2)
>
> **Can the same strategy be applied to VAEs with other priors with explicit distributions? For example, priors of discrete probabilities such as Dirichlet distribution or Softmax computation after Gaussian sampling. Or, exponential families in general.**
>
> Yes, we can use the same strategy for other distributions. However, in the case of discrete random variables, we should utilize a proper MCMC method, for instance:
>
> [1] Nishimura, Akihiko, David B. Dunson, and Jianfeng Lu. "Discontinuous Hamiltonian Monte Carlo for discrete parameters and discontinuous likelihoods." Biometrika 107.2 (2020): 365-380.
>
> In the case of the exponential family of distributions, we can use the HMC we described in the paper. Then, in some very specific cases, we can even take advantage of calculating the posterior in the closed form. However, we do not consider this scenario in the paper since it is quite limiting in terms of applications.

---

> > ### Author Response · Authors · 2022-08-02
> > **Answer to Reviewer d2M4 (2)**
> >
> >
> > **What is the input scale? 0-1 for every experiment?**
> >
> > Yes, inputs were scaled to [0, 1] for all the experiments.
> > We rescale images to [0, 255] only when computing MSE and FID scores.
> >
> > **What is AVAE-SS in Table 3?**
> >
> > This is the VAE variation from [1]. It stands for Self-supervised autoencoding variational autoencoder. They use samples from the trained decoder to “fine-tune” the VAE and increase the smoothness of the encoder, which, in turn, increases the robustness of VAE. The numbers that we report are taken from Table 1 in [1].
> >
> > [1] Cemgil, Taylan, et al. "The autoencoding variational autoencoder." Advances in Neural Information Processing Systems 33 (2020): 15077-15087.
> >
> > **How does the performance change when apply the proposed HMC method to the other VAE variants in Table 3?**
> >
> > We have implemented three variations of VAE: the vanilla version, $\beta$-VAE and $\beta$-TCVAE. In appendix C.2 we report the performance of all these methods with and without the HMC.
> >
> > **In the caption of Table 2: radius 0.1 (top) and 0.2 (bottom), perhaps typos?**
> >
> > Yes, that’s a typo. We’ve fixed it in the updated version of the manuscript.
> >
> > **Typo in Line 332, a dash -.**
> >
> > Thank you for noticing, we’ve updated the manuscript accordingly.

---

> > ### Comment · Reviewer_d2M4 · 2022-08-08
> > **Additional Questions**
> >
> > Dear authors,
> >
> > Thank you for your clarification.
> > Here is a follow-up question: how do the inference time and performance change if the one utilizes other MCMC techniques, for example, the Metropolis–Hastings? (especially, for the NVAE case, if your time allows.)

---

> > > ### Author Response · Authors · 2022-08-08
> > > **Response to follow-up question**
> > >
> > > Dear reviewer,
> > >
> > > Thank you for appreciating our rebuttal.
> > >
> > > There is a variety of MCMC methods that use Matropolis-Hastings correction step. HMC that we are using can be also seen as a special case of the MH, where a new point is proposed by running the numerical integrator (Leapfrog). The main advantage of the method is that it allows us to incorporate the geometry of the target distribution (see Appendix B.1 for the details).
> > >
> > > One can claim that using a cheaper proposal (e.g. random walk) might be beneficial for the inference time. However, there would still be two bottlenecks:
> > >
> > > 1. Since the proposal distribution does not use any information about the target distribution, much more samples would be required.
> > > 2. The acceptance test for each proposed point requires a forward pass through the decoder to compute the density ratio.
> > >
> > > Combining these two points, we believe that using HMC is a reasonable choice in our scenario.

---

> > > > ### Comment · Reviewer_d2M4 · 2022-08-09
> > > > **Thanks**
> > > >
> > > > Dear author,
> > > >
> > > > Thank you for your quick reply. I increased my score to 5, which is in the acceptance region, since my questions are resolved. However, I think that the discussion on the choice of HMC should be included in the main paper somehow.

---

### Author Response · Authors · 2022-08-02
**Comment to all reviewers**

Dear reviewers,
We would like to thank you all for your time and effort in reading and evaluating our paper. We have put a lot of effort into addressing your questions and concerns in the comments below.

We would like to mention that we have added several sections and experiments to the appendix of the paper based on your suggestions (all marked in blue in the updated version of the manuscript). As a result, the numbering of sections and figures has changed (in the appendix only). *In all of the responses, we refer to the updated version of the manuscript.*

---

### Comment · Area_Chair_94wM · 2022-08-09
**Some additional concerns**

Dear Authors

Sorry to dump this on you with so little of the author discussion period left, but I had some additional concerns to those raised by the reviewers and thought it was best to still try and give you a chance to respond to them while that is still potentially possible, even though I appreciate it might not be possible to do so and this point and that any response might need to be rushed.  It is also worth pointing out here that there are also lots of aspects of the paper I like as well (in particular I think the core idea is very neat), but I am just trying to gather as much information as possible before making the final decision.

Thanks

AC

- I am not convinced by the hypothesis that the adversarial attack always results in incorrect reconstructions because we encoder to low probably mass regions under the true posterior.  Specifically, I think this oversimplifies the problem and is a little overgeneralising.  I do agree that this is sensible intuition for a _possible_ failure mechanism, but I do not think it is the only way one can successfully attack a VAE.  In particular, it is possible for $p_{\theta}(x|z)$ to be very peaked in an poor region of reconstructions while still having $p_{\theta}(z|x=x_{true})$ be significant for that $z$.  As such, though a lower “true” posterior density can be indicative of an attack, it is not a necessary condition and I do not think it is appropriate to suggest a rather strong _causal_ hypothesis like this.  I think the intuition here is still useful and the ultimate motivation for the suggested approach (mostly) sound, but it needs rejigging to be more measured and not gloss over the subtleties of what is happening.  Though the paper never makes this clear, I think the key here is really about being able to utilise the encoder to protect the decoder as it is unlikely the attack as exploited both, rather the decoder being an inherently better model than the encoder with some infallible “true posterior” (I would actually also discourage the use of this phrase as the decoder is no more of a “true” model than the encoder, both are being learned and they each have relative merits depending on problem context).  In other words, for me, the key is that randomness of the embeddings means the encoder is generally being attacked more directly than the decoder, so it is better to try and approximate the implied representation defined by the decoder posterior.
- There are some significant missing references with regard to using MCMC methods to achieve tighter ELBOs in a VAE context.  Perhaps the most significant of these is the Hamiltonian Variational Auto-Encoder (Caterini et al, NeurIPS 2018) seems to be very algorithmically similar to the approach being used (albeit with a highly distinct context such as the fact it is deployed during training).  This, and similar papers, should be cited and algorithmic links acknowledged.
- The proof of Lemma 1 is somewhat informal, not quite mathematically complete, and requires additional (mild) assumptions to be made in the Lemma statement about the moments of the log densities.  Specifically, the use of a Taylor expansion is not valid for all g and its required assumptions will indeed be violated at points where p_{\theta}(z|x)->0.  Thus, Eq (16) does not hold uniformly and you cannot immediately assume that the little o term is actually still little o in expectation (as it is actually running of to infinity for some zero-measure region of the input).  In other words, it is actually non-trivial to prove that Eq (21) formally holds from Eq (20) because the expectation of a little o term is not necessarily itself little o without uniform convergence.  I am happy that the high-level result is still essentially correct (other than missing assumptions on moments), but more work is actually required for formal proof.  I would suggest looking at the proof of Theorem 1 in https://arxiv.org/abs/1802.04537 and the proof of Proposition 1 in https://arxiv.org/abs/1705.09279 as these both deal with the omitted issue here by using explicit remainder terms and then showing the expectation of these remainder terms is bounded.  You may even be able to directly leverage the former to do a lot of the heavy lifting for you, as you may be able to simply assert an intermediate result in their proof to make the required jump.  I think it is important that this lemma (and its proof) are updated to make them formally correct, or if this is not possible, downgrading it to a more informal theoretical analysis rather than a formal result (noting I have no issues with the general arguments, just the incorrect presentation of this as a formal result).

---

> ### Comment · Area_Chair_94wM · 2022-08-09
> **Additional concerns part 2**
>
> - The discussion after Lemma 1 is also not really technically correct.  Lemma 1 is an asymptotic result and even then it only really guarantees that the error goes down at a rate quicker than the norm of the attack noise.  Saying that the KL is proportional to the norm of the noise implies a much stronger, non-asymptotic, result.  This claim should thus be removed and a much more measured discussion given.  I would also suggest that this is something which should be empirically directly checked; my experience is that you may well see behaviour quite different than the KL scaling proportionally to the noise for realistic sizes of noise.  The discussion of Theorem 1 is also suspect for the same reason: the final term need only decay faster than the square root of the attack radius for asymptotically small attacks, and may well behave very differently for finitely sized ones.
> - The suggested approach seems to do very poorly on raw downstream accuracy (i.e. adversarial accuracy with zero magnitude) and I think this is a little worrying.  As one of the reviewers alluded to, I also think it means that the claims that “performance on non-attacked inputs is not decreased” is actually true: one does not generally know up front if the input has been attacked so one cannot simply switch the defence mechanism off for certain inputs.  I am actually quite surprised by this drop though and suspect it from training the classifier on the means as you explain in your rebuttal.  I think it is thus important to test this with a classifier trained on samples as well to see how things change and whether this provides a full explanation.
> - Related to the above, the need to move from deterministic to stochastic representations is actually a significant practical limitation that needs acknowledging and ideally more directly investigating, the discussion in the response to bo5D about HMC versus optimisation is important and here relevant here.
> - I’m somewhat suspicious whether the results in Figure 4 are representative.  It does not seem to make sense that the attacker gets worse when they know the defence strategy.  Appendix C.3 offers some useful insight here and I am certainly not trying to suggest any kind of foul play, but I think that there are likely some tangential effects causing some of the apparent differences.  For example, the attacker using MCMC is leading to attacks that are much smaller than their allowed attack radius.  This suggests a failure of the specific optimization algorithm of the attacker (e.g. getting stuck in a local mode), rather than it necessarily actually being harder to attack.  For example, Figure 11 is showing that when the attacker uses MCMC, the attacks are very intuitive to how a human might manually construct an attack and make relatively small changes in pixel space that lead to a change of class.  This is quite a different story than is conveyed in Section 4.3.  My conclusion would be that it is actually very hard to properly control the “strength” of an attack, because hyperparameters and local optimisation mean the attack radius is not actually a reliable measure of the strength of the attack across models.  Thus, though I do not think the results in Section 4.3 are wrong (in the sense that there is no real suggestion of a bug or similar), I do think the current way they are conveyed will lead the reader to misguided conclusions and the section does not properly convey the subtlety of what is going on.

---

> > ### Author Response · Authors · 2022-08-09
> > **Reply to Area Chair**
> >
> > Dear Area Chair,
> >
> > We would like to thank you for thoroughly reading and evaluating our paper. We have put our effort to address your concerns below and update the paper accordingly (changes compared to the latest revision are marked in blue).
> >
> > **I am not convinced by the hypothesis that the adversarial attack always results in incorrect reconstructions because we encoder to low probably mass regions under the true posterior. Specifically, I think this oversimplifies the problem and is a little overgeneralising.**
> >
> > This hypothesis was exactly meant to provide our intuition on why running MCMC may help to alleviate the effect of adversarial attacks. This is the intuition we had from the beginning, so we also wanted to clearly communicate it in the paper. Thanks to the suggestion of the reviewer, we have also added the empirical evidence supporting this hypothesis (currently in Appendix C.1 due to the page limit).
> >
> > Based on your comment, we have updated formulations on lines 50-51 and 138-139 to highlight that this is merely our motivation, but not a necessary condition for each adversarial attack.
> >
> > **Though the paper never makes this clear, I think the key here is really about being able to utilise the encoder to protect the decoder as it is unlikely the attack as exploited both, rather the decoder being an inherently better model than the encoder**
> >
> > We actually tend to think of the method as using the *decoder* to defend the *encoder*. As you correctly noticed, we can use the former to approximate the latent representation even for an attacked point.
> >
> > **rather the decoder being an inherently better model than the encoder with some infallible “true posterior”**
> >
> > We use “true posterior” not to claim that the decoder is better than the encoder, but rather to clearly distinguish it from the variational posterior. The latter is often seen in the VAE literature, of course, and we just wanted to ensure that it is clear what we are referring to. Do you think that writing “exact posterior” instead of the “true posterior” would be a better formulation?
> >
> >
> > **Missing related works**
> >
> > Thank you for pointing to interesting related work! We did have some work on the topics in the background section, namely, Hamiltionan VI and Contrastive Divergence for combining VI and MCMC. We have now added more details on the topic as well as the missing references (see Appendix B.3)
> >
> > **Theoretical Analysis**
> >
> > Thanks a lot for pointing out the missing assumption in Lemma 1. We agree that it should be indeed carefully proven that the little *o* term still behaves as the little *o* under the expectation.
> > In our case the remainder part takes the form (if we consider the Lagrange form of the remainder):
> > \begin{align}
> > R_1(x, x^r, z) = (x - x^r)^T \nabla^2_{x x} g(x + \theta (x - x^r), z ) (x - x^r)^T, \\
> > \theta \in (0, 1)
> > \end{align}
> >
> > Taking the expectation with respect to $z$, we will have to ask $E_z\left[ \nabla^2_{x x} g(x + \theta (x - x^r), z ) \right] $ to be bounded over $z$ for o little property to hold.
> > It is impossible to properly update the proof now due to time constraints, but we will surely do that. In this short time we were given, we have made the following updates to the paper to address your concerns:
> >
> > - Updated the assumption required for the Taylor expansion in Lemma 1 in equation (16).
> >
> > - Updated the discussion after Lemma 1 (lines 160-162) and Theorem 1 (line 193) highlighting that the result only holds for an asymptotically small attack radius.
> >
> > **Drop in the classification accuracy**
> >
> > We agree that performance on the non-attacked points is an important metric. While we observe that the reconstruction error benefits from running HMC, the accuracy of the downstream classifier becomes worse. Unfortunately, we still believe that optimizing for the posterior mode will not solve this problem due to the first argument we mentioned in the HMC vs Optimization discussion: A mode of the $p_{\theta}(z|x^a)$ does not guarantee to coincide with the mode of $q_{\phi}(z|x^a)$. However, we do agree that the paper will benefit a lot if this discussion is added (it is currently briefly mentioned in lines 150-152 and extensively in Appendix B.3, we will add more details to the main body of the camera ready if the paper is accepted)
> >
> > To have empirical evidence that training VAE on samples would (at least partially) solve the problem, we managed to do a small-scale experiment on the MNIST dataset. We use samples from the variational posterior $q(z|x)$ to train a classifier with a test accuracy 0.92. If we run HMC during the evaluation step (without retraining the classifier), the accuracy drops to 0.91. This is a very promising empirical observation, we will update the paper when we compute adversarial accuracy for this classifier.

---

> > > ### Author Response · Authors · 2022-08-09
> > > **Reply to Area Chair (2)**
> > >
> > > **Results in section 4.3 (ablation study)**
> > >
> > > We agree that this is surprising, that the attacker gets weaker when using the defense strategy in attack construction. In our setup, an attacker uses projected gradient descent (due to constraint on the $L_{inf}$ norm of the perturbation) maximizing the $L_2$ distance between two samples obtained by running the HMC in the latent space (see eq. (62 - 64)). After each PGD iteration, the attacker runs a new Markov chain. Therefore, we assume that randomness induced by this procedure makes PGD very ineffective.
> > >
> > > We agree that the gradient information from the decoder should be useful for the attacker. We tried exactly the same pipeline for optimization as in other experiments and did not perform well. However, we do not want to claim it is impossible, but rather that using the same pipeline to construct the attack as we use without the HMC does not work.
> > >
> > > As we discussed at the beginning of Section 5, it might be possible that other strategies to construct adversarial attacks would be much more successful in dealing with the HMC defense. And we believe that it is an exciting direction for future work.
> > >
> > > We are convinced that Figure 11 shows that the attacks are unsuccessful. You are totally right that we see how the attacked point is converted to a different class with a small change in the pixel space. However, this change is perfectly visible (because we only see this result for the large norm of the attack) and does not lead to an unexpected performance of the VAE. We expect something that looks like “9” to be reconstructed and classified as “9” and not as the reference point “4” (here we are referring to the example at the bottom of Figure 11). This is why we conclude in Section 4.3 that it is hard to construct an attack.

---

> > > > ### Comment · Area_Chair_94wM · 2022-08-09
> > > > **Thanks**
> > > >
> > > > Thanks for the follow up and sorry again for giving you this on such late notice.
> > > >
> > > > Your comments all sound very reasonable and this seems like good progress to get everything sorted, I appreciate more time will be needed to do everything fully and I'm pretty impressed what you managed to do this quickly!
> > > >
> > > > I also realized that there was a typo in my earlier comment: I meant that it should be made clear that you are using the decoder to help the encoder as you say you view it yourselves.

---

### Meta-Review · Area_Chair_94wM · 2022-08-30

**Recommendation:** Accept
**Confidence:** Less certain

**Metareview:**

This paper received generally positive reviews, with all reviewers backing acceptance (albeit typically quite weakly) after discussion with the authors.  The paper was praised on a number of fronts such as novelty, potential significance, clarity, and experimental evaluation.  Though there were a few high-level concerns raised about the work by the reviewers (at least not those which were not successfully explained away by the authors), there were some notable lower-level technical concerns, such as the drop in performance for non-adversarial inputs and increased inference time.

My own view of the paper is that the underlying ideas and core contributions of the work are very strong and clearly worthy of acceptance, but there are also quite a large number of technical and lower-level issues that need to be sorted out for publication (as laid out in my comment to the authors).   I do not feel that any of these is individually especially serious, but together they do represent quite a large body of things that would need addressing for the camera ready and one could argue that they would collectively represent too large of an update to accept the work.

On balance, my recommendation is to give the authors the benefit of the doubt and accept the work, as the underlying contribution is clearly solid and there are no fundamental flaws.  Most importantly, I think all of the issues raised, though numerous, are addressable for the camera ready and the authors seem to be already making good progress towards this.  I also feel that the large number of low-level issues raised are partly due to the relatively high level of scrutiny the paper has undergone, rather than being purely a reflection of the number of issues actually present.  I do hope though that the authors take the concerns raised seriously and make appropriate updates for the camera ready.

Additional comments to those given in the previous comment to authors
- The introduction should make it much clearer that you are utilising the decoder to help protect the encoder.  For me, the key to what is going on here is that the encoder is what is used for downstream tasks and is also the thing that is vulnerable to attack because it is applied before the randomness of the embedding.  Using the, more robust, decoder to help protect the encoder is a really neat idea, but I do not think this is currently properly emphasised at present.  Without making this clear I think the approach is quite confusing: the first impression I got when you say that you use MCMC to “fix the latent code” is that you are utilising forbidden information about the original input, rather than utilising the fact we essentially have access to two models.
- The first page of the paper has a lot of grammatical errors and some poorly worded sentences, it needs a few low-level edit passes.  This is not such an issue later in the paper, but there are still some errors later to correct.


**Award:**

No

---

### Decision · Program_Chairs · 2022-09-14

Accept